# Post-Hoc Merging is Not Enough: Many-Shot Model Merging with Loss-Gap Balancing

Kyungjin Im [* 1]  Miru Kim [* 2]  Chanin Eom [* 1]  Minhae Kwon [2]

## Abstract

Model merging has become a practical post-training strategy for building a single multi-task large language model (LLM) by combining multiple task-specialized models. However, most existing approaches rely on post-hoc merging, in which task-specific models are merged only once after training. This one-shot aggregation often suffers from task interference, leading to *information erasure* across individual tasks. In this work, we show that replacing post-hoc merging with an iterative *many-shot merging* protocol is effective in improving multi-task performance. Building on this insight, we propose **METIS**, **M**itigating **E**rasure from **T**ask **I**nterference for **S**table many-shot merging. METIS is a loss-aware many-shot merging method that addresses information erasure in post-hoc merging through task-wise loss-gap weighting and consensus-based masking. Notably, METIS exhibits significant performance improvement on the worst-performing task, effectively mitigating information erasure. `Project Page:` `https://imkyungjin.github.io/METIS/`

## 1. Introduction

LLMs have achieved remarkable success across a wide range of tasks, including natural language understanding [1, 2, 3, 4, 5, 6] and mathematical reasoning [7, 8]. As a result, a central goal in recent LLM research has shifted from excelling at individual tasks toward enabling strong multi-task capability within a single model. However, the massive parameter scale of modern LLMs makes training a task-general model via post-training prohibitively expensive. While LoRA mitigates this cost by updating only a

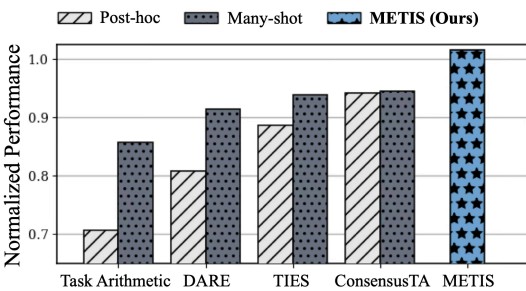

*Figure 1.* Performance comparison of many-shot and post-hoc merging across representative methods using Llama-3.2-3B.

small subset of parameters [9], achieving effective multi-task capability within a single model remains a fundamental challenge.

To address this challenge, *model merging* has emerged as an effective post-training paradigm for constructing multi-task LLMs by aggregating task-specific post-trained models [10, 11, 12, 13, 14, 15]. By reusing independently optimized task models, it provides a scalable alternative to joint multi-task training. However, despite its strong empirical performance, model merging is inherently susceptible to *information erasure* caused by *task interference* [12, 16], which is commonly attributed to *model drift*: task-specific updates push models toward disparate task-optimal regions in parameter space, making naive aggregation prone to overwriting task-specific knowledge [12, 17, 18].

From an optimization and distributed learning perspective, such drift can be mitigated through frequent parameter aggregation, which constrains models within a shared parameter neighborhood and limits task-wise divergence [19, 20, 21]. However, most existing model merging methods rely on a *post-hoc* strategy, performing a single merge after fully post-training task-specific models [12, 14, 16, 22, 23, 24, 25]. This one-shot integration induces abrupt cross-task interference, exacerbating information erasure and ultimately constraining multi-task performance. These observations raise the following question.

*How can we improve the multi-task capability of large language models through model merging, without erasing task-specific knowledge?*

*Equal contribution  [1]Soongsil University, Republic of Korea  [2]Department of Electrical and Computer Engineering, Sungkyunkwan University, Republic of Korea. Correspondence to: Minhae Kwon <minhae.kwon@skku.edu>.

*Proceedings of the 43rd International Conference on Machine Learning*, Seoul, South Korea. PMLR 306, 2026. Copyright 2026 by the author(s).

We argue that moving beyond post-hoc merging toward *many-shot* merging provides a principled answer to this question. In the many-shot merging framework, task-specific models are integrated through a sequence of incremental merging steps. By gradually introducing cross-task interactions, many-shot merging better aligns with the iterative nature of optimization and mitigates the abrupt parameter shifts induced by one-shot aggregation. This integration enables the merged model to adapt to task interactions over time, thereby reducing destructive interference arising from heterogeneous task updates.

Figure 1 provides empirical support for our claim. We investigate whether simply adopting a many-shot merging scheme can already improve upon conventional post-hoc merging. Specifically, we compare the multi-task performance of post-hoc and many-shot merging across representative post-hoc merging baselines, including Task Arithmetic [10], DARE [14], TIES [16], and ConsensusTA [12]. Notably, transitioning from post-hoc to many-shot merging consistently improves multi-task performance across all baseline methods. These results suggest that iterative integration is a key factor in effective multi-task model merging. However, while many-shot merging reduces destructive interference by gradually introducing cross-task interactions, it does not explicitly control how heterogeneous task updates are integrated during the iterative merging process.

Motivated by these findings, we propose **METIS**, a novel merging method that mitigates information erasure under the many-shot merging framework. Rather than naively applying many-shot merging, METIS introduces a task-wise loss-gap-based weighting strategy that balances the contributions of task-specific models, thereby effectively alleviating task interference. In addition, we employ a consensus-based masking mechanism to enhance task localization, building on prior findings [12]. Together, these components enable stable iterative merging and substantially improve multi-task capability while preserving task-specific knowledge.

In this work, we make the following contributions. (i) We theoretically and empirically demonstrate that a simple transition from post-hoc merging to many-shot merging significantly improves the multi-task capability of LLMs. (ii) We propose a novel merging method based on task-wise loss gaps, which effectively mitigates information erasure and is supported by theoretical analysis. (iii) We introduce METIS, a many-shot merging framework that employs consensus-based masking and loss-gap-guided task-aware weighting. We show that the proposed method achieves superior multi-task performance across diverse and representative LLM benchmarks using a single merged model. Additionally, we further verify its robustness against information erasure and the worst-performing task. The notations used in this paper are presented in Appendix A.

## 2. Related Works

**Model Merging.** Model merging aims at integrating multiple fine-tuned models into a single model that retains the multi-task capabilities [22, 26, 27]. Recent studies demonstrate that merging fine-tuned models from different tasks provides better initialization for new tasks [28, 29] and improves out-of-distribution robustness [22, 30, 31, 32]. Model merging also enables multi-task models [10, 13] and supports federated learning [19, 33], continual learning [34], and other settings [35, 36]. This wide range of applications leads to the development of methods that improve beyond simple parameter averaging.

One direction focuses on manipulating task vectors through scaling, pruning, or masking to reduce task interference. Task Arithmetic [10] introduces a scaling factor that controls the magnitude of task vectors, balancing the influence of different tasks while keeping the hyperparameter search tractable with a single shared coefficient. TIES [16] addresses task interference through a three-step pipeline that trims small-magnitude parameters, selects a dominant sign, and merges only aligned weights. DARE [14] applies random dropout to task vectors using Bernoulli masks and rescales the remaining parameters. ConsensusTA [12] further introduces task-specific binary masks and aggregates them via a consensus mechanism to retain parameters agreed upon across tasks. Another direction tackles task interference by decomposing model updates into structured subspaces, often using orthogonalization or singular value decomposition. Iso-C and Iso-CTS [23] propose an isotropic model merging by decomposing parameters into a common subspace shared across tasks and task-specific subspaces, ensuring balanced representation without bias toward dominant tasks. TSV-M [24] leverages SVD to identify principal directions of task updates, reducing interference by aligning merging along dominant singular vectors. Subspace-Boosting [25] emphasizes informative subspaces while suppressing noisy components, effectively boosting performance through subspace reweighting. While these approaches aim to mitigate task interference, they fundamentally operate under a post-hoc merging paradigm, which may still induce abrupt interactions and information loss. This suggests that addressing interference at the level of the merging process itself remains an open challenge.

**Iterative Model Merging Beyond Post-Hoc.** Model merging methods have traditionally focused on combining independently fine-tuned checkpoints to integrate task-specific models. Recent studies show that post-hoc merging may have limitations [18, 37], while iterative merging has been shown to improve reasoning and generalization. Prior work further demonstrates that models leveraging self-generated data and merged over multiple iterations achieve stronger reasoning performance [38]. Iterative merging

has also been shown to help preserve previously acquired knowledge while incorporating new task updates more effectively [21, 39, 40].

From a broader optimization perspective, this paradigm is closely related to distributed and federated learning, where models are updated through repeated aggregation steps rather than a single post-hoc merge [19, 36, 41, 42, 43, 44, 45, 46, 47]. Prior federated learning approaches perform iterative parameter updates across multiple rounds, enabling gradual alignment of heterogeneous task updates and reducing divergence across clients [19, 36]. These studies collectively suggest that iterative integration is a more effective paradigm for handling heterogeneous task updates compared to one-shot merging. The proposed method follows this perspective by explicitly mitigating information erasure while preserving task contributions, enabling more balanced knowledge integration across multiple tasks.

# 3. Many-Shot Merging Improves Multi-task Capability

In this section, we evaluate the effectiveness of many-shot merging by first introducing the system setting, followed by theoretical analysis and empirical validation.

## 3.1. Many-Shot Merging Framework

In this scheme, we consider total $T$ models, each associated with a unique task $\tau \in \mathcal{T}$, where $\mathcal{T} = \{1, \ldots, T\}$ denotes the task set. Each model is trained on its own dataset $\mathcal{D}_\tau$ corresponding to task $\tau$. All models start from the same pre-trained parameter initialization $\Theta^0$ and are then independently trained using their respective task-specific losses $\mathcal{L}_\tau(\cdot)$.

$$\theta_\tau^{\mathrm{r}} \leftarrow \Theta^{\mathrm{r-1}} - \eta \nabla \mathcal{L}_\tau \left( \Theta^{\mathrm{r-1}} \right) \tag{1}$$

In (1), $\eta$ means learning rate and $\mathrm{r} \in \{1, \ldots, \mathrm{R}\}$ denotes the update round. After the local update, each model computes a task vector $\boldsymbol{v}_\tau$, defined as the difference between the locally updated parameters at round $\mathrm{r}$ and the initial model $\Theta^0$, i.e., $\boldsymbol{v}_\tau^{\mathrm{r}} = \theta_\tau^{\mathrm{r}} - \Theta^0$. The merging phase then proceeds using these task vectors as follows.

$$\Theta^{\mathrm{r}} = \mathcal{M} \left( \boldsymbol{v}_1^{\mathrm{r}}, \ldots \boldsymbol{v}_\tau^{\mathrm{r}}, \ldots, \boldsymbol{v}_T^{\mathrm{r}} \right) \tag{2}$$

Herein, $\mathcal{M}(\cdot)$ is a merging operator, which can be instantiated by various merging methods [10, 12, 14, 16]. After many-shot merging over $\mathrm{R}$ rounds, all models share an identical merged model $\Theta^{\mathrm{R}}$. An illustrative example of many-shot merging and post-hoc merging is presented in Figure 2.

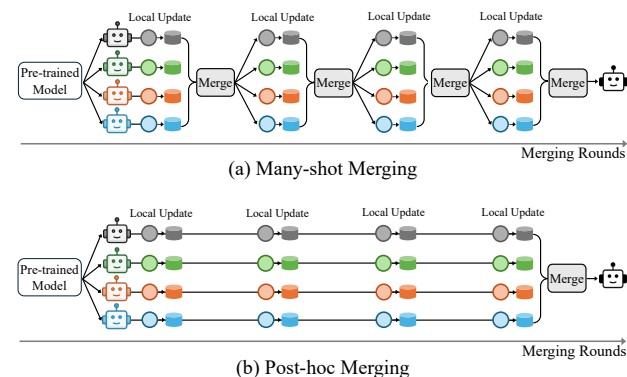

(a) Many-shot Merging

(b) Post-hoc Merging

*Figure 2.* Overview of the merging frameworks: (a) many-shot merging and (b) post-hoc merging, under the same number of local updates.

## 3.2. Reducing Multi-Task Loss with Many-Shot Merging

We observe that many-shot merging improves multi-task performance over post-hoc merging. In this subsection, we provide theoretical and empirical support for this observation using the following definition of the multi-task loss.

**Definition 3.1** (Multi-task Loss $\mathcal{E}(\Theta^{\mathrm{r}})$)**.** Given the merged parameters at round $r$, denoted by $\Theta^{\mathrm{r}}$, the multi-task loss $\mathcal{E}(\Theta^{\mathrm{r}})$ is defined as the average of task-specific losses.

$$\mathcal{E}(\Theta^{\mathrm{r}}) = \frac{1}{T} \sum_{\tau=1}^{T} \mathcal{L}_\tau \left( \Theta^{\mathrm{r}} \right) \tag{3}$$

Since the multi-task loss $\mathcal{E}(\Theta^{\mathrm{r}})$ aggregates task-specific losses, a lower value of $\mathcal{E}(\Theta^{\mathrm{r}})$ indicates stronger multi-task generalization ability. From this perspective, we provide the following theoretical analysis supporting the improved multi-task capability of the many-shot merging strategy.

**Theorem 3.2** (Multi-task Loss Reduction)**.** *Suppose each task loss $\mathcal{L}_\tau(\cdot)$ is $L$-smooth, and the learning rate satisfies $\eta \leq 1/L$. Let $\bar{\Theta}^R$ denote the post-hoc merged model, and let $\Theta^R$ be the many-shot merged model obtained after $\mathrm{R}$ rounds of local updates. Then, the following inequality is always satisfied under the condition $\Delta(\mathcal{E}, R) + \frac{L}{2} \Delta(\xi, R) \leq 0$.*

$$\mathcal{E}\left(\Theta^R\right) - \mathcal{E}\left(\bar{\Theta}^R\right) \leq 0 \tag{4}$$

*Proof sketch.* Since each task loss $\mathcal{L}_\tau(\cdot)$ is $L$-smooth, the aggregated multi-task objective is also $L$-smooth [48, 49]. By the $L$-smoothness inequality, $\mathcal{E}(\Theta^{\mathrm{R}}) - \mathcal{E}(\bar{\Theta}^{\mathrm{R}})$ is written as follows.

$$
\begin{aligned}
&\mathcal{E}(\Theta^{\mathrm{R}}) - \mathcal{E}(\bar{\Theta}^{\mathrm{R}}) \\
&\leq \frac{1}{T} \sum_{\tau=1}^{T} \mathcal{E}(\theta_\tau^{\mathrm{R}}) + \frac{L}{2} \xi^{\mathrm{R}} - \left( \frac{1}{T} \sum_{\tau=1}^{T} \mathcal{E}(\bar{\theta}_\tau^{\mathrm{R}}) + \frac{L}{2} \bar{\xi}^{\mathrm{R}} \right)
\end{aligned} \tag{5}
$$

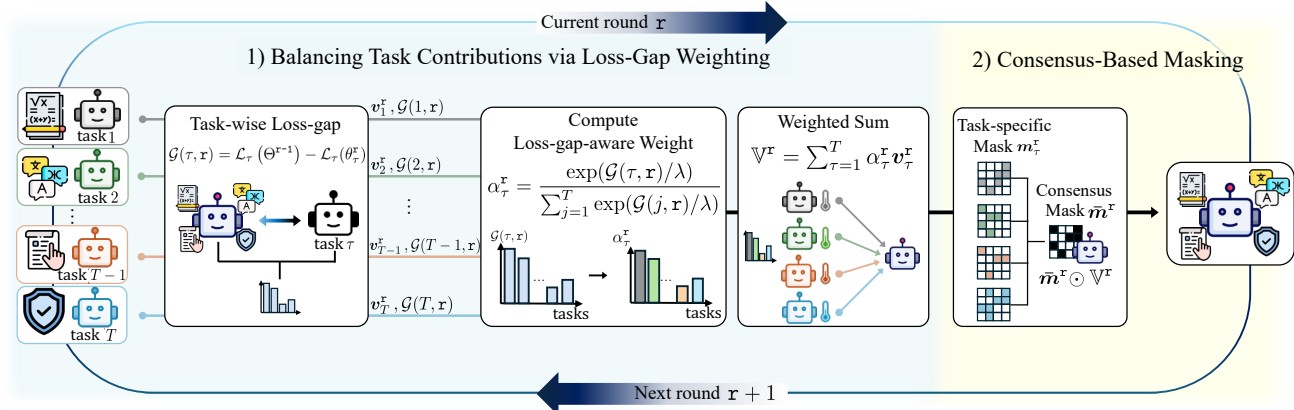

*Figure 3.* Overall framework of the proposed solution. The method balances task contributions via loss-gap-based weighting and enhances task localization through consensus-based masking within the many-shot merging framework.

Then, (5) can be written as follows.

$$\frac{1}{T} \sum_{\tau=1}^{T} \left( \mathcal{E}(\theta_\tau^{\text{R}}) - \mathcal{E}(\bar{\theta}_\tau^{\text{R}}) \right) + \frac{L}{2} \left( \xi^{\text{R}} - \bar{\xi}^{\text{R}} \right)$$

$$= \Delta(\mathcal{E}, \text{R}) + \frac{L}{2}\Delta(\xi, \text{R}) \leq 0 \qquad (6)$$

The inequality in (6) always holds because $\Delta(\mathcal{E}, \text{R}) + \frac{L}{2}\Delta(\xi, \text{R}) \leq 0$ is guaranteed by the stated condition.[1] The complete proof is provided in Appendix B.1. □

**Empirical Support.** Table 1 provides empirical support for Theorem 3.2. Specifically, we compare representative post-hoc merging methods, including Task Arithmetic [10], DARE [14], TIES [16], and ConsensusTA [12]. Across all methods, simply transitioning from post-hoc to many-shot merging consistently reduces the multi-task loss. Beyond loss reduction, we further examine whether many-shot merging improves multi-task capability. To this end, we adopt the normalized performance measure introduced in [12, 37]. Consistent with the multi-task loss results, we observe notable gains in normalized performance across all post-hoc merging methods. These results clearly demonstrate that many-shot merging improves multi-task capability.

Although we have verified that many-shot merging improves multi-task capability over post-hoc merging, recent studies [12, 16] show that model merging remains inherently vulnerable to *task interference*, which can cause *information erasure* and degrade performance on certain tasks. In the next section, we present a novel merging method that further mitigates *information erasure* under the many-shot merging framework.

---

[1] It is noteworthy that this condition readily holds. We provide empirical results in Appendix B.2 showing that representative merging methods consistently satisfy this condition.

*Table 1.* Multi-task loss and performance results for post-hoc and many-shot merging on Llama-3.2-3B.

| Method | Multi-task Loss ($\downarrow$) | | | Performance ($\uparrow$) | | |
|---|---|---|---|---|---|---|
| | Post-hoc | | Many-shot | Post-hoc | | Many-shot |
| Task Arithmetic | 2.97 | $\rightarrow$ | **2.00** | 0.706 | $\rightarrow$ | **0.857** |
| DARE | 1.93 | $\rightarrow$ | **1.67** | 0.807 | $\rightarrow$ | **0.914** |
| TIES | 1.81 | $\rightarrow$ | **1.66** | 0.883 | $\rightarrow$ | **0.938** |
| ConsensusTA | 1.83 | $\rightarrow$ | **1.49** | 0.942 | $\rightarrow$ | **0.945** |

## 4. METIS: Mitigating Information Erasure with Task-Wise Loss-Aware Consensus

In this section, we propose **METIS**, which effectively addresses the problem of *information erasure* in many-shot merging [12]. Our method employs task-wise loss-gap weighting to rebalance task contributions and consensus-based masking to localize compatible parameter updates, thereby reducing task interference. Figure 3 provides an overview of the proposed framework.

### 4.1. Balancing Task Contributions via Loss-Gap Weighting

Since we adopt a many-shot merging framework, information erasure for a specific task can be compensated in subsequent merging rounds by assigning a higher weight to the corresponding task model. Motivated by this insight, we define the task-wise loss-gap, which quantifies the extent of information erasure incurred in the most recent merging round.

**Definition 4.1** (Task-wise Loss Gap $\mathcal{G}(\tau, r)$). Let $\Theta^{r-1}$ be the merged model at round $r-1$ and $\theta_\tau^r$ be the locally adapted model for task $\tau$ at round $r$. Then, the task-wise loss gap $\mathcal{G}(\tau, r)$ for task $\tau$ at round $r$ is defined as follows.

$$\mathcal{G}(\tau, r) = \mathcal{L}_\tau \left( \Theta^{r-1} \right) - \mathcal{L}_\tau(\theta_\tau^r) \qquad (7)$$

According to (7) in Definition 4.1, a larger $\mathcal{G}(\tau, \mathtt{r})$ indicates that the most recently merged model $\Theta^{\mathtt{r}-1}$ fits task $\tau$ worse than its locally adapted counterpart, and thus that task $\tau$ should receive a larger merging weight in the current round.

**Loss-Gap-Aware Task Vector Aggregation.** Based on the task-aware loss gap defined in (7), we balance task contributions during merging. To this end, we derive the loss-gap-aware merged task vector $\mathbb{V}^{\mathtt{r}}$ as follows.

$$\mathbb{V}^{\mathtt{r}} = \sum_{\tau=1}^{T} \underbrace{\left( \frac{\exp(\mathcal{G}(\tau, \mathtt{r})/\lambda)}{\sum_{j=1}^{T} \exp\left(\mathcal{G}(j, \mathtt{r})/\lambda\right)} \right)}_{\text{loss-gap-aware weight } \alpha_{\tau}^{\mathtt{r}}} \boldsymbol{v}_{\tau}^{\mathtt{r}} \quad (8)$$

Herein, $\lambda \in \mathbb{R}^{+}$ controls the sharpness of the reweighting. With the loss-gap-aware weight $\alpha_{\tau}^{\mathtt{r}}$, the merged task vector $\mathbb{V}^{\mathtt{r}}$ places greater emphasis on task-specific models that are underrepresented in the most recent merging round. More specifically, tasks with more severely erased information are assigned larger values of $\alpha_{\tau}^{\mathtt{r}}$, thereby contributing more strongly to the current aggregation, whereas tasks that are already well captured by the merged model receive smaller weights. Such loss-gap-aware aggregation is only feasible within the many-shot merging framework, as computing the task-wise loss gap $\mathcal{G}(\tau, \mathtt{r})$ requires access to the previously merged model. By integrating many-shot merging with loss-gap-aware weighting, METIS can further mitigate the information-erasure problem in model merging.

In the remainder of this subsection, we provide theoretical evidence that loss-gap-aware task vector weighting mitigates information erasure. To this end, we compare the worst-performing task loss, $\mathcal{L}_{\tilde{\tau}}(\Theta)$, after merging under average task vector aggregation and loss-gap-aware weighting.

**Theorem 4.2** (Information Erasure Robustness). *Let $\Theta^{\dagger}$ and $\Theta^{\circ}$ denote the merged models obtained via loss-gap-aware aggregation using (8) and mean aggregation respectively. Assume that the worst-task loss function $\mathcal{L}_{\tilde{\tau}}(\Theta)$ is $L$-smooth. Then, the following inequality holds.*

$$\mathbb{E}\left[\mathcal{L}_{\tilde{\tau}}(\Theta^{\dagger})\right] \leq \mathbb{E}\left[\mathcal{L}_{\tilde{\tau}}(\Theta^{\circ})\right]. \quad (9)$$

*Proof sketch.* From the merging rule in (2) and the definition of $\alpha_{\tau}$ in (8), each merging process is defined as follows.

$$\Theta^{\circ} = \Theta + \sum_{\tau=1}^{T} \frac{1}{T} \boldsymbol{v}_{\tau}, \quad \Theta^{\dagger} = \Theta + \sum_{\tau=1}^{T} \alpha_{\tau} \boldsymbol{v}_{\tau} \quad (10)$$

$\mathcal{L}_{\tilde{\tau}}(\cdot)$ is $L$-smooth, the expected difference between $\mathcal{L}_{\tilde{\tau}}(\Theta^{\dagger})$ and $\mathcal{L}_{\tilde{\tau}}(\Theta^{\circ})$ can be bounded using the descent lemma [50, 51] as follows, where $d^{\circ} = \sum_{\tau} \frac{1}{T} \boldsymbol{v}_{\tau}$ and $d^{\dagger} = \sum_{\tau} \alpha_{\tau} \boldsymbol{v}_{\tau}$.

$$\mathbb{E}\left[\mathcal{L}_{\tilde{\tau}}(\Theta^{\dagger}) - \mathcal{L}_{\tilde{\tau}}(\Theta^{\circ})\right] \leq \mathbb{E}\left[\langle g_{\tilde{\tau}}, d^{\dagger} - d^{\circ}\rangle\right]$$
$$+ \frac{L}{2}\left(\mathbb{E}\|d^{\dagger}\|^{2} - \mathbb{E}\|d^{\circ}\|^{2}\right) \quad (11)$$

---

**Algorithm 1** METIS
1: **Input:** Pre-trained model $\Theta^{0}$, task set $\mathcal{T} = \{1, \dots, T\}$, datasets $\{\mathcal{D}_{\tau}\}_{\tau \in \mathcal{T}}$, rounds $\mathtt{R}$
2: **Output:** Merged model $\Theta^{\mathtt{R}}$
3: Initialize $\Theta^{0}$
4: **for** $\mathtt{r} = 1$ to $\mathtt{R}$ **do**
5:     **for** each task $\tau \in \mathcal{T}$ **do**
6:         Local update: $\theta_{\tau}^{\mathtt{r}} \leftarrow \Theta^{\mathtt{r}-1} - \eta \nabla \mathcal{L}_{\tau}(\Theta^{\mathtt{r}-1})$
7:         Task vector: $\boldsymbol{v}_{\tau}^{\mathtt{r}} \leftarrow \theta_{\tau}^{\mathtt{r}} - \Theta^{0}$
8:         Loss gap: $\mathcal{G}(\tau, \mathtt{r}) \leftarrow \mathcal{L}_{\tau}(\Theta^{\mathtt{r}-1}) - \mathcal{L}_{\tau}(\theta_{\tau}^{\mathtt{r}})$
9:     **end for**
10:     Loss-gap-aware weights $\{\alpha_{\tau}^{\mathtt{r}}\}$ by (8)
11:     Weighted task vector: $\mathbb{V}^{\mathtt{r}} \leftarrow \sum_{\tau=1}^{T} \alpha_{\tau}^{\mathtt{r}} \boldsymbol{v}_{\tau}^{\mathtt{r}}$
12:     **for** each task $\tau \in \mathcal{T}$ **do**
13:         Task-specific mask $\boldsymbol{m}_{\tau}^{\mathtt{r}}$ via (12)
14:     **end for**
15:     Consensus mask $\bar{\boldsymbol{m}}^{\mathtt{r}}$ via (13)
16:     Merged update: $\Theta^{\mathtt{r}} \leftarrow \Theta^{0} + \beta^{\mathtt{r}}(\bar{\boldsymbol{m}}^{\mathtt{r}} \odot \mathbb{V}^{\mathtt{r}})$
17: **end for**

---

Under a standard bounded-interference condition, loss-gap reweighting assigns larger weight to underperforming tasks, implying $\mathbb{E}\langle g_{\tilde{\tau}}, d^{\dagger}\rangle \leq \mathbb{E}\langle g_{\tilde{\tau}}, d^{\circ}\rangle$. Furthermore, when update magnitudes are controlled, the quadratic difference term is non-positive or negligible [52]. Therefore, the right-hand side is non-positive, which gives $\mathbb{E}[\mathcal{L}_{\tilde{\tau}}(\Theta^{\dagger})] \leq \mathbb{E}[\mathcal{L}_{\tilde{\tau}}(\Theta^{\circ})]$. The complete proof for Theorem 4.2 is provided in Appendix B.3. $\square$

Theorem 4.2 establishes that the proposed loss-gap-aware weighting further reduces the loss of the worst-performing task. This indicates that severely erased task information is better recovered after applying our weighting method. When integrated into the many-shot merging framework, this mechanism promotes balanced knowledge integration across tasks, simultaneously mitigating information erasure and improving the average multi-task performance.

### 4.2. Consensus-Based Masking for Localization

Beyond directly updating the model using the loss-gap-aware task vector $\mathbb{V}^{\mathtt{r}}$, we additionally incorporate a consensus-based masking mechanism for localization [12]. In this study, we jointly employ both task-specific masks and a consensus mask.

**Task-Specific Mask ($\boldsymbol{m}_{\tau}^{\mathtt{r}}$).** We consider a binary task-specific masking vector $\boldsymbol{m}_{\tau}^{\mathtt{r}}$ that determines which elements of the merged task vector $v_{i}^{\mathtt{r}} \in \mathbb{V}^{\mathtt{r}}$ degrade task-specific information. Specifically, the mask on the $i$-th element $m_{\tau,i}^{\mathtt{r}} \in \boldsymbol{m}_{\tau}^{\mathtt{r}}$ is activated only when the weighted update of the $i$-th task is sufficiently aligned with the remainder of the

merged update, as defined below.

$$m_{\tau,i}^{\text{r}} = \mathbb{I}\Big(\alpha_\tau^{\text{r}}\big|v_{\tau,i}^{\text{r}}\big| \geq \delta\,\big|v_i^{\text{r}} - \alpha_\tau^{\text{r}}v_{\tau,i}^{\text{r}}\big|\Big) \qquad (12)$$

In (12), $\mathbb{I}(\cdot)$ denotes the indicator function and $\delta$ controls the strictness of the masking threshold for task $\tau$. Under this masking scheme, $\boldsymbol{m}_\tau^{\text{r}}$ retains only the coordinates where the contribution of task $\tau$ is not dominated by conflicting contributions from other tasks.

**Consensus Mask ($\bar{\boldsymbol{m}}^{\text{r}}$).** With the task-specific masks $\boldsymbol{m}_\tau^{\text{r}}$, we derive a consensus masking vector $\bar{\boldsymbol{m}}^{\text{r}}$. Each element of this vector is set to one only when at least $k$ task-specific models agree to accept the $i$-th element of the merged task vector $v_i^{\text{r}}$. The consensus mask is computed from the task-specific masks as follows.

$$\bar{m}_i^{\text{r}} = \mathbb{I}\Big(\sum_{\tau=1}^{T} m_{\tau,i}^{\text{r}} \geq k\Big) \qquad (13)$$

Using the consensus mask in (13), the final merged parameters are derived as follows.

$$\Theta^{\text{r}} \leftarrow \Theta^0 + \beta^{\text{r}}\Big(\bar{\boldsymbol{m}}^{\text{r}} \odot \mathbb{V}^{\text{r}}\Big) \qquad (14)$$

Here, $\beta^{\text{r}} \in \mathbb{R}^+$ is a scaling factor that controls the overall magnitude of the merged update [10, 12, 16].

# 5. Experiments

We evaluate the performance of METIS through extensive experiments. In this section, we present the experimental setup, performance comparisons, and a wide range of analyses of the proposed method.

## 5.1. Experimental Setup

**Task Datasets and Pre-Trained Models.** We consider four task categories: 1) instruction-following, 2) mathematical reasoning, 3) multilingual understanding, and 4) safety. For each category, we construct a task-specific fine-tuning set by uniformly sampling 1,000 training instances. Specifically, we use TULU-3 Persona Instruction-Following [53] for instruction-following, DART-Math [54] and Numina-MathTIR [55] for mathematical reasoning, Aya [56] for multilingual understanding, and WildGuardMix [57] together with WildJailbreak [58] for safety. We evaluate on four base LLMs spanning multiple model families and scales: Gemma-2-2B [59], Llama-3.2-3B, Llama-3.1-8B [60], and Qwen-3-4B [61]. To ensure a fair comparison between post-hoc and many-shot merging, we match the total number of task-specific updates across settings and fix the number of merging rounds to R = 5.

**Baseline Methods.** We compare the proposed approach against representative model merging methods, including Task Arithmetic [10], DARE [14], TIES [16], and ConsensusTA [12]. All baselines are evaluated under both post-hoc and many-shot merging frameworks to isolate the effect of iterative merging. For each method, scaling factors and method-specific hyperparameters are selected based on validation performance following standard practice [12, 37].

**Evaluation Settings.** To evaluate our approaches in the model merging setting, we follow the MergeBench [37] protocol, a standardized benchmark for comparing merging methods. Instruction-following is evaluated on IFEval [62] using prompt-level accuracy, and mathematical reasoning on GSM8K [7] using exact match under an 8-shot chain-of-thought setting. Multilingual understanding is measured on M-MMLU, M-ARC, and M-HellaSwag [63], and safety on XSTest [64] using accuracy. To account for varying task difficulty, we report normalized performance, defined in Appendix C.2. The raw performance values are reported in Appendix D.7.

## 5.2. Performance Comparisons

In this subsection, we compare the multi-task capability of the proposed method with representative post-hoc merging approaches. In addition, we evaluate many-shot variants of each baseline for a comprehensive comparison.

**Post-Hoc Merging Comparison.** Table 2 provides a performance comparison with post-hoc merging methods. Across all backbone model settings, the proposed method consistently achieves the highest average normalized performance, indicating superior overall multi-task capability. In contrast, several baseline methods exhibit substantial variability across backbone models, with their minimum average performance dropping sharply as the backbone changes. Although TIES demonstrates relatively more stable behavior across different backbones, its average performance remains consistently below that of the proposed method in all settings. These results suggest that the proposed many-shot-based methodology yields stronger and more reliable multi-task capability than post-hoc-based baselines.

**Many-Shot Merging Comparison.** The same table (Table 2) compares the proposed method with many-shot variants of existing post-hoc approaches. Extending post-hoc methods to the many-shot setting consistently improves their performance compared to post-hoc merging, demonstrating the benefit of many-shot merging in multi-task settings. Notably, the proposed method continues to achieve the highest average normalized performance across most backbone models. The advantage of the proposed method is best illustrated by comparison with ConsensusTA. Al-

*Table 2.* Category-level normalized performance across backbone models on instruction following, mathematical reasoning, multilingual understanding, and safety tasks, including both individual category scores and their average. The best results are shown in bold, and the second-best results are underlined.

**(a) Gemma-2-2B**

| | Method | Avg. | Inst. | Math | Multi. | Safety |
|---|---|---|---|---|---|---|
| Post-Hoc | Task Arithmetic | 0.521 | 0.250 | 0.080 | 0.692 | 0.721 |
| Post-Hoc | DARE | 0.663 | 0.375 | 0.420 | 0.765 | 0.885 |
| Post-Hoc | TIES | 0.726 | 0.275 | 0.440 | **0.940** | 0.820 |
| Post-Hoc | ConsensusTA | 0.752 | 0.400 | 0.600 | 0.827 | 1.033 |
| Many-Shot | Task Arithmetic | 0.715 | 0.500 | 0.520 | 0.898 | 0.574 |
| Many-Shot | DARE | 0.701 | 0.350 | 0.520 | 0.839 | 0.820 |
| Many-Shot | TIES | 0.776 | 0.400 | 0.540 | 0.840 | 1.197 |
| Many-Shot | ConsensusTA | 0.791 | 0.450 | **0.660** | 0.814 | 1.197 |
| Many-Shot | **METIS (Ours)** | **0.800** | **0.525** | 0.620 | 0.815 | **1.213** |

**(b) Llama-3.2-3B**

| | Method | Avg. | Inst. | Math | Multi. | Safety |
|---|---|---|---|---|---|---|
| Post-Hoc | Task Arithmetic | 0.706 | 0.583 | 0.385 | 0.715 | 1.122 |
| Post-Hoc | DARE | 0.807 | 0.542 | 0.615 | 0.854 | 1.122 |
| Post-Hoc | TIES | 0.883 | 0.375 | 0.897 | **1.082** | 0.776 |
| Post-Hoc | ConsensusTA | 0.942 | 0.875 | 0.641 | 0.958 | **1.265** |
| Many-Shot | Task Arithmetic | 0.857 | 0.375 | 0.744 | 1.048 | 0.878 |
| Many-Shot | DARE | 0.914 | 0.792 | 0.846 | 0.955 | 0.980 |
| Many-Shot | TIES | 0.938 | 0.792 | **0.923** | 0.924 | 1.143 |
| Many-Shot | ConsensusTA | 0.945 | **0.917** | 0.872 | 0.907 | 1.163 |
| Many-Shot | **METIS (Ours)** | **1.015** | **0.917** | 0.872 | 1.018 | 1.245 |

**(c) Llama-3.1-8B**

| | Method | Avg. | Inst. | Math | Multi. | Safety |
|---|---|---|---|---|---|---|
| Post-Hoc | Task Arithmetic | 0.704 | 0.390 | 0.525 | 0.796 | 0.919 |
| Post-Hoc | DARE | 0.838 | **0.634** | 0.695 | 0.885 | **1.047** |
| Post-Hoc | TIES | 0.852 | 0.390 | 1.017 | **1.002** | 0.698 |
| Post-Hoc | ConsensusTA | 0.694 | 0.293 | 0.458 | 0.812 | 0.977 |
| Many-Shot | Task Arithmetic | 0.785 | 0.244 | 0.966 | 0.996 | 0.512 |
| Many-Shot | DARE | 0.849 | 0.439 | 0.966 | 0.965 | 0.791 |
| Many-Shot | TIES | 0.902 | 0.390 | 1.017 | **1.002** | 1.000 |
| Many-Shot | ConsensusTA | 0.898 | 0.512 | 1.085 | 0.943 | 0.965 |
| Many-Shot | **METIS (Ours)** | **0.935** | 0.585 | **1.136** | 0.972 | 0.977 |

**(d) Qwen-3-4B**

| | Method | Avg. | Inst. | Math | Multi. | Safety |
|---|---|---|---|---|---|---|
| Post-Hoc | Task Arithmetic | 1.048 | 0.909 | 0.750 | 1.036 | 1.517 |
| Post-Hoc | DARE | 0.986 | 0.818 | 0.614 | 0.994 | 1.500 |
| Post-Hoc | TIES | 1.108 | 1.136 | 0.886 | 1.029 | 1.534 |
| Post-Hoc | ConsensusTA | 1.067 | 1.000 | 0.750 | 1.040 | 1.534 |
| Many-Shot | Task Arithmetic | 1.084 | 0.682 | **0.977** | 1.069 | 1.638 |
| Many-Shot | DARE | 1.154 | 1.136 | 0.886 | **1.087** | 1.638 |
| Many-Shot | TIES | 1.087 | 1.000 | 0.818 | 1.044 | 1.569 |
| Many-Shot | ConsensusTA | 1.128 | 1.182 | 0.830 | 1.052 | 1.603 |
| Many-Shot | **METIS (Ours)** | **1.180** | **1.318** | 0.920 | 1.062 | **1.655** |

though both operate within the many-shot framework and use masking-based integration, the proposed method consistently achieves higher performance across backbone models, demonstrating that loss-gap-aware task-wise aggregation enhances multi-task capability beyond masking alone.

## 5.3. Merging Robustness Analyses

In this section, we examine the robustness of the proposed method across three aspects: (1) information erasure, (2) out-of-domain generalization, and (3) worst-case robustness.

**Robustness to Information Erasure.** Figure 4 compares task-level performance across benchmarks for multiple backbone LLMs. Across all backbones, the proposed method maintains balanced performance across tasks without severe degradation on any individual benchmark. In contrast, baseline methods often achieve strong performance on specific tasks while exhibiting noticeable drops on others, indicating imbalanced task integration. These results suggest that explicitly rebalancing task contributions via task-wise loss-gap aggregation enables more stable alignment of heterogeneous task objectives and effectively mitigates information erasure during iterative merging.

**Robustness to Worst-Performing Task.** A robust multi-task model should perform well across all tasks, including

the most challenging ones. To assess worst-case behavior, we evaluate the capability of merged models on their worst-performing task. Table 3 reports both the average performance and the performance on the worst-performing task across four backbone models. Notably, the proposed method achieves the highest performance on the worst-performing task while also attaining the best average performance. Moreover, it exhibits the smallest performance gap between the average and worst-performing tasks among all baselines, indicating more balanced task-wise capability. These results demonstrate that the proposed method improves overall multi-task robustness across tasks.

**Robustness to Pre-trained Knowledge Forgetting.** Beyond solving in-domain tasks during merging, we further examine how well the merged LLM retains pre-trained knowledge in Table 4. Specifically, we evaluate pre-trained and merged models on CoQA [65] and PubMedQA [66], benchmarks on which the pre-trained models are known to perform well. Interestingly, the proposed method consistently remains closest to the pre-trained model's performance compared to other merging approaches. This result demonstrates that the proposed merging strategy more effectively preserves pre-trained knowledge and transferable representations under multi-task integration, which is critical for reliable generalization to unseen tasks.

*Table 3.* Average and worst-performing task results compared with post-hoc merging methods across backbone models. The value in parentheses indicates the drop from the average performance.

| Model ($\rightarrow$) | Gemma-2-2B | | Llama-3.2-3B | | Model ($\rightarrow$) | Llama-3.1-8B | | Qwen-3-4B | |
|---|---|---|---|---|---|---|---|---|---|
| Method ($\downarrow$) | Avg. | Worst-task | Avg. | Worst-task | Method ($\downarrow$) | Avg. | Worst-task | Avg. | Worst-task |
| Task Arithmetic | 0.521 | 0.080 (-0.44) | 0.706 | 0.385 (-0.32) | Task Arithmetic | 0.704 | 0.390 (-0.31) | 1.048 | 0.750 (-0.30) |
| DARE | 0.663 | 0.375 (-0.29) | 0.807 | 0.542 (-0.27) | DARE | 0.838 | **0.634** (-0.20) | 0.986 | 0.614 (-0.37) |
| TIES | 0.726 | 0.275 (-0.45) | 0.883 | 0.375 (-0.51) | TIES | 0.852 | 0.390 (-0.46) | 1.108 | 0.886 (-0.22) |
| ConsensusTA | 0.752 | 0.400 (-0.35) | 0.942 | 0.641 (-0.30) | ConsensusTA | 0.694 | 0.293 (-0.40) | 1.067 | 0.750 (-0.32) |
| **METIS (Ours)** | **0.800** | **0.525** (-0.28) | **1.015** | **0.872** (-0.14) | **METIS (Ours)** | **0.935** | 0.585 (-0.35) | **1.180** | **0.920** (-0.26) |

*Table 4.* Pre-trained knowledge performance on CoQA (exact match) and PubMedQA (accuracy) across backbone models, compared with representative post-hoc merging methods. Gray-shaded rows denote the performance of the corresponding pre-trained backbone without task-specific fine-tuning.

| Model ($\rightarrow$) | Gemma-2-2B | | Llama-3.2-3B | | Model ($\rightarrow$) | Llama-3.1-8B | | Qwen-3-4B | |
|---|---|---|---|---|---|---|---|---|---|
| Method ($\downarrow$) | CoQA | PubMedQA | CoQA | PubMedQA | Method ($\downarrow$) | CoQA | PubMedQA | CoQA | PubMedQA |
| Pre-trained | 0.695 | 0.732 | 0.659 | 0.750 | Pre-trained | 0.697 | 0.766 | 0.683 | 0.746 |
| Task Arithmetic | 0.420 | 0.558 | 0.548 | 0.552 | Task Arithmetic | 0.573 | 0.554 | 0.473 | 0.688 |
| DARE | 0.561 | 0.622 | 0.552 | 0.556 | DARE | 0.687 | 0.654 | 0.639 | 0.708 |
| TIES | 0.705 | **0.720** | 0.662 | 0.552 | TIES | 0.687 | 0.574 | 0.661 | **0.728** |
| ConsensusTA | 0.645 | 0.640 | 0.583 | 0.556 | ConsensusTA | 0.577 | 0.554 | 0.521 | 0.688 |
| **METIS (Ours)** | **0.714** | 0.718 | **0.670** | **0.596** | **METIS (Ours)** | **0.698** | **0.728** | **0.684** | **0.728** |

*Table 5.* Normalized performance of baseline methods under the many-shot setting on Llama-3.2-3B across 7, 8, and 11 tasks.

| Method (Many-shot Setting) | 7 Tasks | 8 Tasks | 11 Tasks |
|---|---|---|---|
| Task Arithmetic | 0.997 | 0.970 | 1.021 |
| DARE | 1.006 | 0.975 | 0.812 |
| TIES | 0.871 | 0.822 | 0.976 |
| ConsensusTA | 0.927 | 0.823 | 0.873 |
| **METIS (Ours)** | **1.015** | **0.993** | **1.036** |

**Robustness to Number of Tasks.** We further examine whether the proposed method remains effective as the number of merged tasks increases. We evaluate our method on three benchmark sets studied in previous works: a 7-task NLP benchmark [16], an 8-task question-answering benchmark [67], and their union, resulting in 11 tasks overall. Table 5 reports normalized performance under the many-shot setting on Llama-3.2-3B with 7, 8, and 11 tasks. METIS consistently achieves the highest performance across all task counts. While several baselines exhibit noticeable performance fluctuations as task diversity increases, METIS maintains strong performance, demonstrating that loss-gap-aware aggregation scales effectively to more challenging multi-task merging settings.

### 5.4. Ablation Study

To quantify the contribution of each design choice, we conduct a component-wise ablation study in which individual components are selectively disabled while all other configu-

rations are held fixed. Specifically, we evaluate the effects of (1) many-shot merging, (2) loss-gap-aware weighting, and (3) consensus-based masking on Llama-3.2-3B.

Figure 5 shows that combining all proposed components achieves the best performance, demonstrating their complementary roles. Disabling any single component consistently degrades performance, indicating that each contributes meaningfully. In particular, removing many-shot merging results in the largest drop, highlighting its central role in mitigating information erasure across merging rounds. Excluding loss-gap-aware weighting also causes clear degradation, confirming its importance in balancing task contributions and reducing destructive interference. Although removing consensus masking still yields strong performance, it remains inferior to the full design.

### 5.5. Sensitivity Analyses

In this subsection, we present sensitivity analyses on the task masking threshold and backbone model size. Additional results are provided in Appendix D.3.

**Task Masking Threshold.** Figure 6(a) presents a sensitivity analysis of the proposed method with respect to the task masking threshold $\delta$ in (12) on Llama-3.1-8B. As $\delta$ increases, the performance of the proposed method slightly decreases, but no abrupt degradation is observed. Moreover, even under higher threshold values, the proposed method consistently surpasses the best performance of ex-

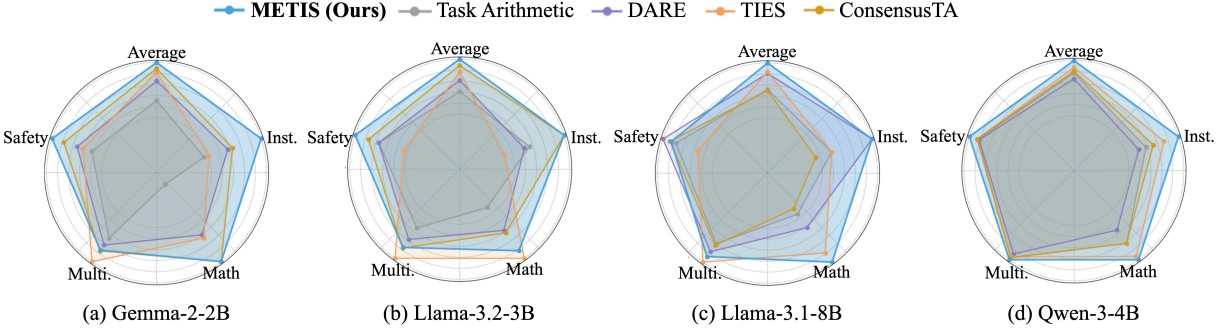

(a) Gemma-2-2B     (b) Llama-3.2-3B     (c) Llama-3.1-8B     (d) Qwen-3-4B

*Figure 4.* Normalized performance comparison between the proposed method and post-hoc merging baselines across four task categories (instruction-following, mathematical reasoning, multilingual understanding, and safety). Scores are normalized for each benchmark, with the best-performing method set to 1.0.

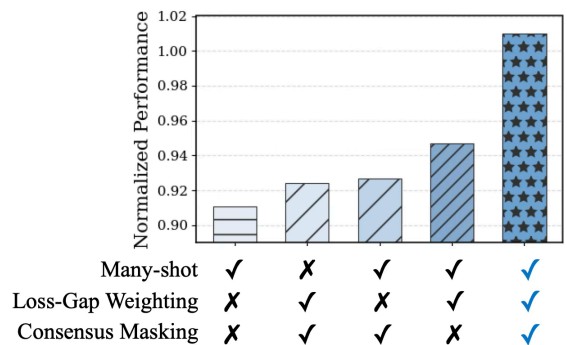

*Figure 5.* Component-wise ablation analysis of the proposed method for Llama-3.2-3B.

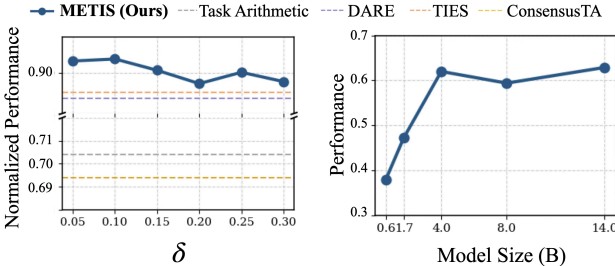

(a) Sensitivity to Task Masking Threshold    (b) Scaling with Model Size

*Figure 6.* Sensitivity analysis of the proposed method. (a) Sensitivity to the task masking threshold $\delta$ on Llama-3.1-8B. (b) Performance scaling with model size across Qwen-3-0.6B, Qwen-3-1.7B, Qwen-3-4B, Qwen-3-8B, and Qwen-3-14B. Dashed horizontal lines indicate post-hoc merging baselines.

isting merging methods. These results indicate that the proposed masking mechanism is robust to the choice of $\delta$ within a reasonable range.

**Model Size.** Figure 6(b) reports normalized performance across backbone model sizes ranging from Qwen-3-0.6B to Qwen-3-14B [61]. As model size increases, the performance of all methods generally improves, reflecting increased representational capacity. The proposed method follows this trend across model scales, indicating favorable scalability with respect to model size.

## 6. Conclusion

In this work, we revisited model merging from the perspective of iterative multi-task optimization and showed that replacing post-hoc merging with a many-shot protocol consistently improves multi-task capability. Building on this insight, we proposed METIS, a loss-aware many-shot merging method that stabilizes iterative integration through task-wise loss-gap weighting and consensus-based masking. Extensive experiments across diverse tasks, model families, and scales demonstrate that METIS achieves stronger and more balanced multi-task performance while mitigating information erasure and preserving out-of-domain general-

ization. These results suggest that progressive, loss-aware merging provides a principled and scalable foundation for reliable multi-task deployment of large language models.

## Impact Statement

This work contributes to the development of more reliable and scalable multi-task LLMs by improving the stability of post-training model merging. By mitigating task interference and reducing information erasure, the proposed method can help practitioners deploy a single merged model that maintains balanced performance across diverse tasks, potentially lowering computational costs and energy consumption compared to training separate models or performing joint multi-task training. While the proposed solution greatly enhances multi-task capability, the many-shot merging framework incurs additional computational overhead due to repeated merging steps. This cost is confined to the training phase and remains modest in practice, as only a small number of updates (around five) are required. Nevertheless, it can be further mitigated by adopting more robust merging strategies within a reduced LoRA parameter space.

## Acknowledgment

This work was supported by the National Research Foundation of Korea (NRF) grant funded by the Korean government (MSIT) under Grant (RS-2023-00278812, RS-2025-02214082), the Institute of Information & Communications Technology Planning & Evaluation (IITP) grant funded by the Korea government (MSIT) (No. RS-2026-25519475), and Korea Institutes of Police Technology (KIPoT) funded by the Korean National Police Agency & Korea Customs Service (RS-2026-25536784).
Kyungjin Im contributed to project conceptualization, algorithm development, manuscript drafting, primary experiments, rebuttal experiments, camera-ready preparation, presentation video preparation, and poster presentation. Miru Kim contributed to project conceptualization, algorithm development, rebuttal writing, camera-ready preparation, and poster presentation. Chanin Eom contributed to manuscript rewriting and theoretical development, including theorem formulation. Minhae Kwon conceived and led the project, secured funding, supervised the research, guided the algorithmic and experimental development, provided extensive manuscript revision and feedback, supervised the rebuttal and camera-ready preparation processes, and advised the conference presentation preparation.

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

# Appendix

# A. Notation

*Table 6.* Notation Table

| Notation | Description |
|---|---|
| $\mathcal{T} = \{1, \dots, T\}$ | Set of tasks, where $T$ denotes the total number of tasks |
| $\tau \in \mathcal{T}$ | Task index |
| $\mathcal{D}_\tau$ | Dataset corresponding to task $\tau$ |
| $\Theta^0$ | Shared pre-trained model parameters used as initialization |
| $\Theta^r$ | Merged model parameters at merging round $r$ |
| $\theta_\tau^r$ | Locally adapted model for task $\tau$ at round $r$ |
| $\mathcal{L}_\tau(\cdot)$ | Task-specific loss function for task $\tau$ |
| $\boldsymbol{v}_\tau^r$ | Task vector for task $\tau$ at round $r$, defined as $\theta_\tau^r - \Theta^0$ |
| $\mathcal{M}(\cdot)$ | Generic merging operator applied to task vectors |
| $\mathcal{E}(\Theta^r)$ | Multi-task loss, defined as the average of task-specific losses |
| $\mathcal{G}(\tau, r)$ | Task-wise loss gap for task $\tau$ at round $r$ |
| $\lambda$ | Temperature parameter controlling the sharpness of loss-gap-aware reweighting |
| $\alpha_\tau^r$ | Loss-gap-aware merging weight for task $\tau$ at round $r$ |
| $\mathbb{V}^r$ | Loss-gap-aware aggregated task vector at round $r$ |
| $\boldsymbol{m}_\tau^r$ | Binary task-specific mask for task $\tau$ at round $r$ |
| $m_{\tau,i}^r$ | Mask value for the $i$-th parameter of task $\tau$ at round $r$ |
| $\delta$ | Threshold parameter controlling the strictness of task-specific masking |
| $\bar{\boldsymbol{m}}^r$ | Consensus mask at round $r$ |
| $k$ | Minimum number of tasks required to activate a parameter in the consensus mask |
| $\odot$ | Element-wise (Hadamard) product |
| $\mathbb{I}(\cdot)$ | Indicator function |
| $R$ | Total number of many-shot merging rounds |
| $\beta^r$ | Scaling factor at round $r$ controlling the magnitude of the merged update |

# B. Theoretical Analysis

### B.1. Proof of Theorem 3.2

*Proof.* Recall that the multi-task loss is defined as

$$\mathcal{E}(\Theta) = \frac{1}{T} \sum_{\tau=1}^{T} \mathcal{L}_\tau(\Theta).$$

Since each task loss $\mathcal{L}_\tau$ is $L$-smooth, their average $\mathcal{E}$ is also $L$-smooth. Hence, for any $x, y$,

$$\mathcal{E}(y) \leq \mathcal{E}(x) + \langle \nabla \mathcal{E}(x), y - x \rangle + \frac{L}{2} \|y - x\|^2.$$

Let $\{\theta_\tau^{\mathrm{R}}\}_{\tau=1}^{T}$ be the task-specific models used to construct the many-shot merged model

$$\Theta^{\mathrm{R}} = \frac{1}{T} \sum_{\tau=1}^{T} \theta_\tau^{\mathrm{R}},$$

and let $\{\bar{\theta}_\tau^{\mathrm{R}}\}_{\tau=1}^{T}$ be those used to construct the post-hoc merged model

$$\bar{\Theta}^{\mathrm{R}} = \frac{1}{T} \sum_{\tau=1}^{T} \bar{\theta}_\tau^{\mathrm{R}}.$$

Define the within-round dispersions

$$\xi^{\mathrm{R}} = \frac{1}{T} \sum_{\tau=1}^{T} \|\theta_\tau^{\mathrm{R}} - \Theta^{\mathrm{R}}\|^2, \qquad \bar{\xi}^{\mathrm{R}} = \frac{1}{T} \sum_{\tau=1}^{T} \|\bar{\theta}_\tau^{\mathrm{R}} - \bar{\Theta}^{\mathrm{R}}\|^2.$$

Applying the smoothness inequality of $\mathcal{E}$ at $(x, y) = (\Theta^{\mathrm{R}}, \theta_\tau^{\mathrm{R}})$ yields

$$\mathcal{E}(\theta_\tau^{\mathrm{R}}) \leq \mathcal{E}(\Theta^{\mathrm{R}}) + \langle \nabla \mathcal{E}(\Theta^{\mathrm{R}}), \theta_\tau^{\mathrm{R}} - \Theta^{\mathrm{R}} \rangle + \frac{L}{2} \|\theta_\tau^{\mathrm{R}} - \Theta^{\mathrm{R}}\|^2.$$

Rearranging and averaging over $\tau$ eliminates the linear term due to $\frac{1}{T} \sum_\tau (\theta_\tau^{\mathrm{R}} - \Theta^{\mathrm{R}}) = 0$, giving

$$\mathcal{E}(\Theta^{\mathrm{R}}) \leq \frac{1}{T} \sum_{\tau=1}^{T} \mathcal{E}(\theta_\tau^{\mathrm{R}}) + \frac{L}{2} \xi^{\mathrm{R}}.$$

The same argument applied to $\bar{\Theta}^{\mathrm{R}}$ yields

$$\mathcal{E}(\bar{\Theta}^{\mathrm{R}}) \leq \frac{1}{T} \sum_{\tau=1}^{T} \mathcal{E}(\bar{\theta}_\tau^{\mathrm{R}}) + \frac{L}{2} \bar{\xi}^{\mathrm{R}}.$$

Subtracting the two bounds gives

$$\mathcal{E}(\Theta^{\mathrm{R}}) - \mathcal{E}(\bar{\Theta}^{\mathrm{R}}) \leq \frac{1}{T} \sum_{\tau=1}^{T} \left( \mathcal{E}(\theta_\tau^{\mathrm{R}}) - \mathcal{E}(\bar{\theta}_\tau^{\mathrm{R}}) \right) + \frac{L}{2} \left( \xi^{\mathrm{R}} - \bar{\xi}^{\mathrm{R}} \right).$$

By definition, this right-hand side equals $\Delta(\mathcal{E}, \mathrm{R}) + \frac{L}{2} \Delta(\xi, \mathrm{R})$, which is non-positive under the stated condition. Therefore,

$$\mathcal{E}(\Theta^{\mathrm{R}}) \leq \mathcal{E}(\bar{\Theta}^{\mathrm{R}}).$$

$\square$

**B.2. Empirical Results Under the Condition of Theorem 3.2**

We examine whether the condition of Theorem 3.2 is readily satisfied. The condition is as follows.

$$\Delta(\mathcal{E}, \text{R}) + \frac{L}{2}\Delta(\xi, \text{R}) \leq 0 \tag{15}$$

Table 7 shows that the condition in (15) is commonly satisfied across representative merging algorithms. This result supports the claim that the condition required for the theoretical analysis in Theorem 3.2 is not restrictive, but rather is naturally satisfied in practice.

*Table 7.* Empirical evaluation of the condition in (15) required by Theorem 3.2.

| Algorithm | $\Delta(\mathcal{E}, \mathbf{R})$ | $\frac{L}{2}\Delta(\xi, \mathbf{R})$ | $\Delta(\mathcal{E}, \mathbf{R}) + \frac{L}{2}\Delta(\xi, \mathbf{R})$ |
|---|---|---|---|
| Task Arithmetic | -0.0947 | -0.0002 | **-0.0949** |
| DARE | -0.1517 | -0.0005 | **-0.1522** |
| TIES | -0.1948 | -0.0001 | **-0.1949** |
| ConsensusTA | -0.2391 | -0.0001 | **-0.2392** |

**B.3. Proof of Theorem 4.2**

*Proof.* Let $\Theta$ denote the shared pre-merge parameters at a given round, and define

$$d^\circ = \sum_{\tau=1}^{T} \frac{1}{T}\boldsymbol{v}_\tau, \qquad d^\dagger = \sum_{\tau=1}^{T} \alpha_\tau \boldsymbol{v}_\tau, \quad \text{where } \alpha_\tau \geq 0, \ \sum_{\tau=1}^{T} \alpha_\tau = 1.$$

The corresponding merged models are

$$\Theta^\circ = \Theta + d^\circ, \qquad \Theta^\dagger = \Theta + d^\dagger.$$

Let $g_{\tilde{\tau}} = \nabla\mathcal{L}_{\tilde{\tau}}(\Theta)$.

Since $\mathcal{L}_{\tilde{\tau}}$ is $L$-smooth, the descent lemma gives

$$\mathcal{L}_{\tilde{\tau}}(\Theta + d) \leq \mathcal{L}_{\tilde{\tau}}(\Theta) + \langle g_{\tilde{\tau}}, d\rangle + \frac{L}{2}\|d\|^2 \quad \text{for any } d.$$

Applying this inequality with $d = d^\dagger$ and $d = d^\circ$ and subtracting yields

$$\mathcal{L}_{\tilde{\tau}}(\Theta^\dagger) - \mathcal{L}_{\tilde{\tau}}(\Theta^\circ) \leq \langle g_{\tilde{\tau}}, d^\dagger - d^\circ\rangle + \frac{L}{2}\big(\|d^\dagger\|^2 - \|d^\circ\|^2\big).$$

Taking expectations gives

$$\mathbb{E}\big[\mathcal{L}_{\tilde{\tau}}(\Theta^\dagger) - \mathcal{L}_{\tilde{\tau}}(\Theta^\circ)\big] \leq \mathbb{E}\langle g_{\tilde{\tau}}, d^\dagger\rangle - \mathbb{E}\langle g_{\tilde{\tau}}, d^\circ\rangle + \frac{L}{2}\big(\mathbb{E}\|d^\dagger\|^2 - \mathbb{E}\|d^\circ\|^2\big).$$

Under the standard bounded-interference and magnitude-control conditions, the loss-gap-aware weighting ensures

$$\mathbb{E}\langle g_{\tilde{\tau}}, d^\dagger\rangle \leq \mathbb{E}\langle g_{\tilde{\tau}}, d^\circ\rangle, \qquad \mathbb{E}\|d^\dagger\|^2 \leq \mathbb{E}\|d^\circ\|^2.$$

Hence the right-hand side is non-positive, implying

$$\mathbb{E}\big[\mathcal{L}_{\tilde{\tau}}(\Theta^\dagger)\big] \leq \mathbb{E}\big[\mathcal{L}_{\tilde{\tau}}(\Theta^\circ)\big].$$

$\square$

## C. Experimental Detail

All experiments were conducted using the system configuration summarized in Table 8.

*Table 8.* System Specification

| | |
|---|---|
| **CPU** | AMD Ryzen Threadripper PRO 9975WX |
| **GPU** | 4× NVIDIA RTX PRO 6000 Blackwell |
| **RAM** | 128 GB |
| **SSD** | 2× 4 TB |

### C.1. Training Setup

We perform a many-shot merging framework with a total of $R = 5$ merging rounds. At each round, task-specific adaptation is performed for a 1 epoch using a batch size of 8. We employ the AdamW optimizer with an initial learning rate of $3 \times 10^{-4}$ with a learning rate scheduler. For parameter-efficient fine-tuning, we utilize LoRA adapters with rank $r = 16$ and scaling factor $\alpha = 16$, applied to the query, key, value, and output projection matrices.

### C.2. Normalized Performance Metric

Given the varying difficulty levels across tasks, we report normalized performance using the following metric [12, 37].

$$\frac{1}{n} \sum_{i=1}^{n} \frac{\text{perf}_{\text{merged}}^{(i)}}{\text{perf}_{\text{finetuned}}^{(i)}} \tag{16}$$

Herein, $\text{perf}_{\text{merged}}^{(i)}$ and $\text{perf}_{\text{finetuned}}^{(i)}$ are the performance of the merged and individually fine-tuned models on task $i$.

### C.3. Hyperparameter Selection

**Loss-gap Temperature** $\lambda$. The temperature parameter $\lambda$ controls the sharpness of the loss-gap-aware reweighting in (8). We tune $\lambda$ over the set $\{1, 2, 3, 4\}$ and select the optimal value based on validation performance.

**Task-specific Masking Threshold** $\delta$. The parameter $\delta$ regulates the sparsity–coverage trade-off in the task-specific mask defined in (12). We explore $\delta \in \{0.05, 0.1, 0.15, 0.2, 0.25, 0.3\}$ for all tasks and choose the best configuration via validation.

**Consensus Threshold** $k$. The consensus threshold $k$ specifies the minimum number of tasks that must agree in order to activate a parameter in the consensus mask, as defined in (13). We tune $k$ over $\{1, 2, 3\}$.

**Scaling Factor** $\beta^{\text{r}}$. We additionally introduce a scaling factor $\beta^{\text{r}}$, as used in prior work [10, 12, 16], defined in (14) to modulate the magnitude of the aggregated task vector during merging. The scaling factor is tuned over the range $\beta^{\text{r}} \in [1.0, 2.0]$.

# D. Additional Experimental Result

## D.1. Comparison with Recent Model Merging and Federated Learning Baselines

*Table 9.* Normalized performance of baseline methods on Llama-3.2-3B across four tasks. We compare all recent state-of-the-art merging baselines, federated/iterative approaches, data mixing, and conventional baselines.

| Category | Method | Avg. | Instruction | Math | Multilingual Understanding | | | Safety |
|---|---|---|---|---|---|---|---|---|
| | | | IFEval | GSM8K | M-MMLU | M-ARC | M-HellaSwag | XSTest |
| State-of-the-Art (+Many-shot) | Iso-C | 0.941 | 0.792 | **1.026** | 1.018 | 0.969 | 1.005 | 0.837 |
| | Iso-CTS | 0.920 | 0.958 | 0.897 | 1.012 | 0.988 | **1.014** | 0.653 |
| | TSV-M | 0.953 | 0.750 | 0.821 | 0.964 | 0.994 | 1.005 | 1.184 |
| | TA + Subspace Boosting | 0.885 | 0.458 | 0.744 | 1.229 | 0.957 | 0.963 | 0.959 |
| Iterative / FL | FedMerge | 0.928 | 0.875 | 0.923 | 1.012 | 0.969 | 0.991 | 0.796 |
| | q-FedAvg | 0.858 | 0.875 | 0.846 | 0.916 | 0.988 | 0.688 | 0.837 |
| | ColD Fusion | 0.911 | **0.917** | 0.923 | 0.837 | 0.957 | 0.977 | 0.857 |
| Data-Mixing | TULU 3 | 0.884 | 0.542 | 0.821 | 0.934 | 0.914 | 0.991 | 1.102 |
| Conventional Method (+Many-shot) | Task Arithmetic | 0.857 | 0.375 | 0.744 | **1.235** | 0.957 | 0.953 | 0.878 |
| | DARE | 0.914 | 0.792 | 0.846 | 0.904 | 0.957 | 1.005 | 0.980 |
| | TIES | 0.938 | 0.792 | 0.923 | 0.813 | 0.969 | 0.991 | 1.143 |
| | ConsensusTA | 0.945 | **0.917** | 0.872 | 0.825 | 0.914 | 0.981 | 1.163 |
| **METIS (Ours)** | | **1.015** | **0.917** | 0.872 | 0.910 | **1.155** | 0.991 | **1.245** |

Table 9 presents an extended comparison on Llama-3.2-3B across four task categories, including recent state-of-the-art merging methods, federated and iterative optimization approaches, data-mixing strategies, and conventional baselines. METIS achieves the best overall normalized performance, demonstrating strong multi-task capability across heterogeneous baselines and confirming the effectiveness of loss-gap-aware task balancing and consensus-based masking in the many-shot merging setting.

## D.2. Robustness to Information Erasure

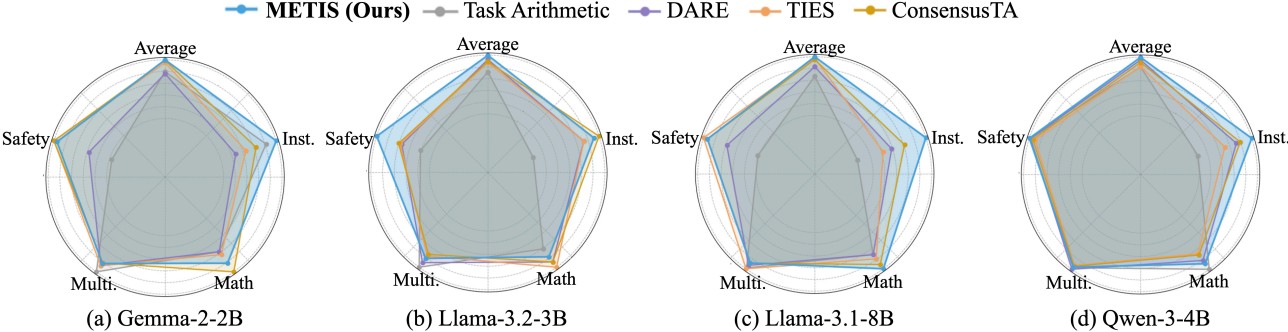

*Figure 7.* Normalized performance comparison between the proposed method and many-shot merging baselines across four task categories (instruction-following, mathematical reasoning, multilingual understanding, and safety). Scores are normalized for each benchmark, with the best-performing method set to 1.0.

Figure 7 reports normalized performance comparisons between the proposed method and many-shot merging baselines across four task categories: instruction-following, mathematical reasoning, multilingual understanding, and safety. These results indicate that the proposed method maintains balanced performance across task categories, demonstrating robustness to information erasure under many-shot merging.

## D.3. Sensitivity Analysis

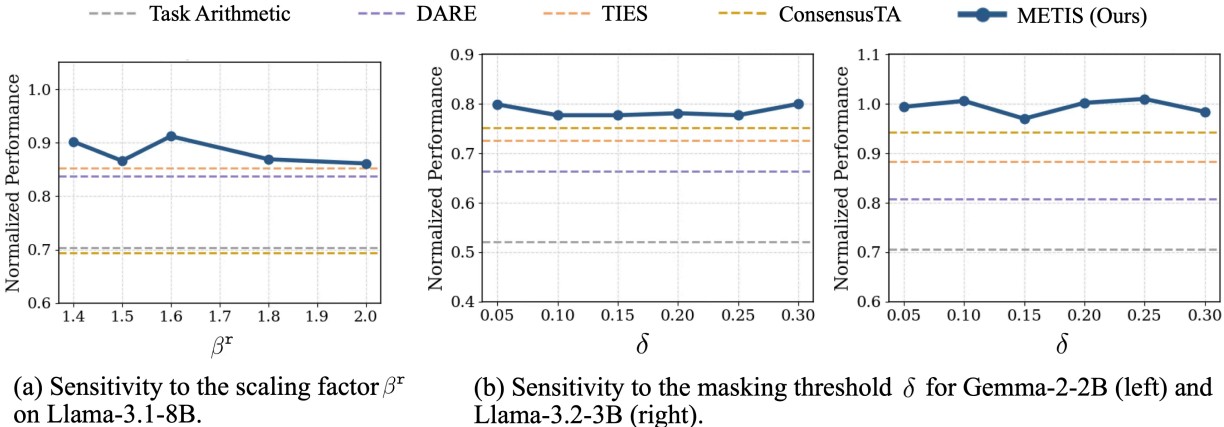

(a) Sensitivity to the scaling factor $\beta^{\mathbf{r}}$ on Llama-3.1-8B.

(b) Sensitivity to the masking threshold $\delta$ for Gemma-2-2B (left) and Llama-3.2-3B (right).

*Figure 8.* Sensitivity analysis of the proposed method. (a) Sensitivity to the scaling factor $\beta^r$ on Llama-3.1-8B. (b) Sensitivity to the masking threshold $\delta$ on Gemma-2-2B and Llama-3.2-3B. Dashed horizontal lines indicate the performance of representative post-hoc merging baselines.

**Sensitivity to the Scaling Factor $\beta^{\mathbf{r}}$.** Figure 8(a) reports the performance trends of the proposed method as the effective scaling magnitude of $\beta^r$ is varied on Llama-3.1-8B, where the x-axis corresponds to representative $\beta^r$ values selected to progressively approach $1.0$ as the merging rounds increase under a round-dependent schedule. The results show that the proposed method remains stable across a reasonable range of $\beta^r$ values, indicating robustness to the scaling magnitude used during merging.

**Sensitivity to the Masking Threshold $\delta$.** Figure 8(b) presents a sensitivity analysis of the proposed method with respect to the masking threshold $\delta$ defined in (12), evaluated on Gemma-2-2B and Llama-3.2-3B. Across different $\delta$ values, the proposed method consistently outperforms representative post-hoc merging baselines. Moreover, the performance remains stable under varying masking thresholds, indicating robustness to the choice of $\delta$ in the proposed masking mechanism.

## D.4. Multi-task Loss Comparison

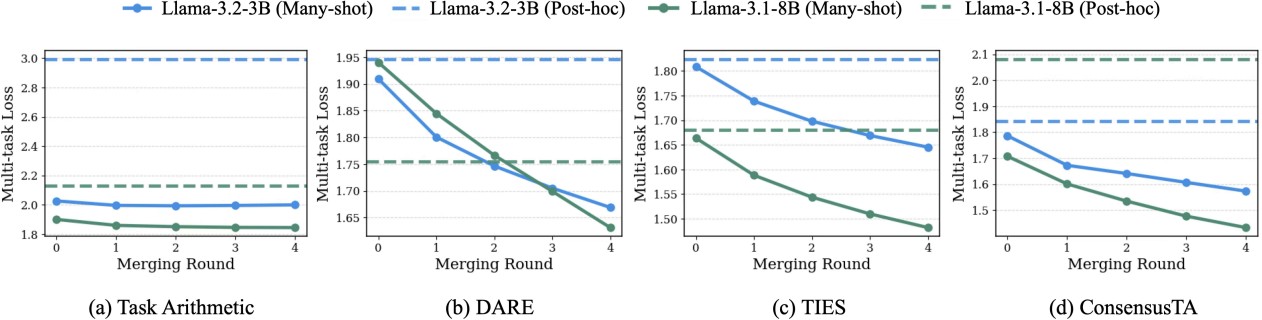

(a) Task Arithmetic      (b) DARE      (c) TIES      (d) ConsensusTA

*Figure 9.* Comparison of post-hoc and many-shot merging dynamics across representative baselines. Many-shot merging consistently reduces multi-task loss over merging rounds, while post-hoc merging corresponds to a single integration step.

Figure 9 compares representative post-hoc merging methods, including Task Arithmetic, DARE, TIES, and ConsensusTA, under post-hoc and many-shot merging settings. Results are reported for Llama-3.2-3B and Llama-3.1-8B. Across all methods, many-shot merging Strategy consistently reduces the

multi-task loss. This empirical observation is consistent with the theoretical analysis Theorem 3.2 regarding iterative merging in multi-task settings.

### D.5. Task-wise Loss-Gap Convergence

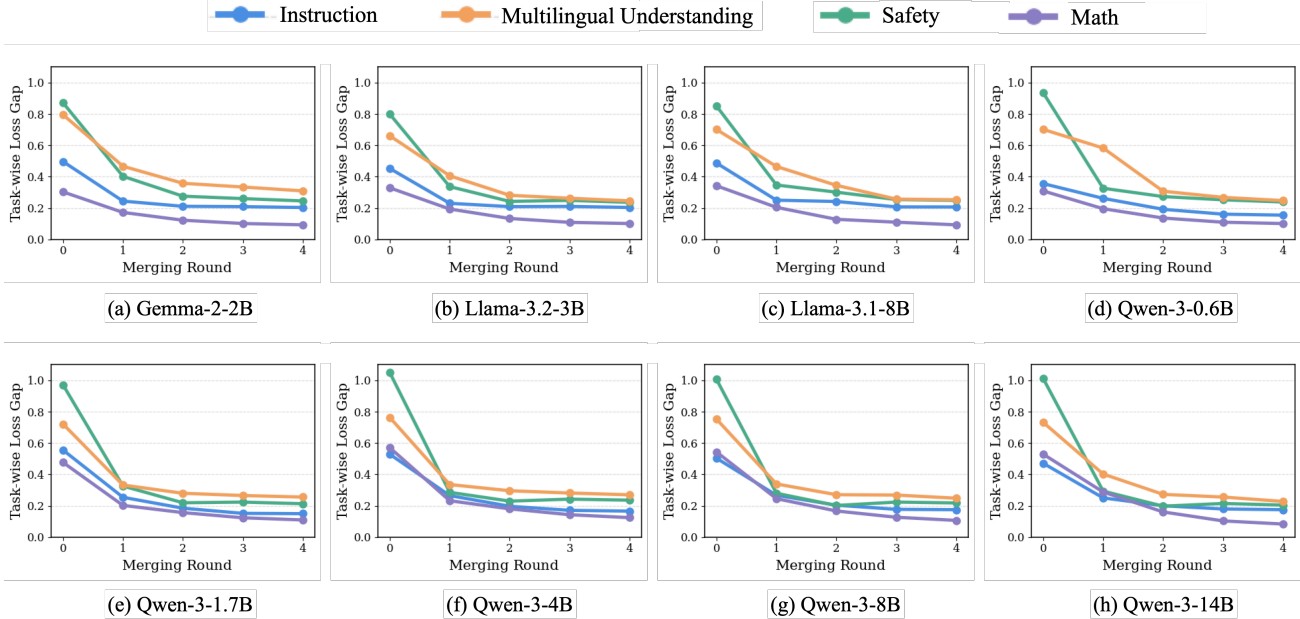

*Figure 10.* Task-wise loss-gap convergence across merging rounds for different base models. Loss gaps consistently decrease, indicating progressive mitigation of information erasure during many-shot merging.

Figure 10 reports the evolution of task-wise loss gaps defined in (7) across merging rounds for multiple base models and task categories. As merging progresses, task-wise loss gaps generally decrease, indicating that discrepancies between the merged model and task-specific models are reduced over rounds. Tasks that exhibit larger loss gaps in earlier rounds show more noticeable reductions in later rounds, which is consistent with the use of loss-gap-aware task weighting.

### D.6. Computational Complexity Analysis

The proposed method achieves the best overall and worst-task performance while maintaining competitive efficiency. Table 10 compares average performance, worst-performing task performance, and runtime across representative many-shot merging methods. While our method incurs slightly higher

*Table 10.* Comparison of performance and computational efficiency under the many-shot setting.

| Method (+Many-shot Setting) | Average | Worst task | Runtime(s) |
|---|---|---|---|
| Task Arithmetic | 0.857 | 0.375 | **883.48** |
| DARE | 0.914 | 0.792 | 874.89 |
| TIES | 0.938 | 0.792 | 900.58 |
| ConsensusTA | 0.945 | 0.825 | 904.79 |
| Iso-C | 0.941 | 0.792 | 1050.18 |
| Iso-CTS | 0.920 | 0.653 | 1225.03 |
| TA+Subspace Boosting | 0.885 | 0.458 | 1252.42 |
| TSV-M | 0.953 | 0.750 | 3219.07 |
| **METIS (Ours)** | **1.015** | **0.872** | 918.53 |

runtime than standard baselines such as Task Arithmetic, DARE, TIES, and ConsensusTA, the overhead remains moderate. Moreover, it is substantially more efficient than stronger baselines such as TSV-M and Iso-CTS. The proposed method achieves the best average and worst-task performance, demonstrating a strong efficiency–performance trade-off.

### D.7. Full Numeric Results

We report the full numeric results of multi-task performance for all merging algorithms in Tables 11, 12, 13, and 14. All results in this subsection correspond to unnormalized performance scores.

*Table 11.* Gemma-2-2B per-task and average results.

| Method | Many-shot | Avg. | Instruction | Math | Multilingual Understanding | | | Safety |
|---|---|---|---|---|---|---|---|---|
| | | | IFEval | GSM8K | M-MMLU | M-ARC | M-HellaSwag | XSTest |
| Task Arithmetic | ✗ | 0.263 | 0.100 | 0.040 | 0.285 | 0.298 | 0.415 | 0.440 |
| | ✓ | 0.351 | 0.200 | 0.260 | 0.345 | 0.410 | 0.543 | 0.350 |
| DARE | ✗ | 0.334 | 0.150 | 0.210 | 0.315 | 0.318 | 0.473 | 0.540 |
| | ✓ | 0.353 | 0.140 | 0.260 | 0.300 | 0.398 | 0.520 | 0.500 |
| TIES | ✗ | 0.365 | 0.110 | 0.220 | 0.365 | 0.430 | 0.563 | 0.500 |
| | ✓ | 0.397 | 0.160 | 0.270 | 0.295 | 0.393 | 0.535 | 0.730 |
| ConsensusTA | ✗ | 0.382 | 0.160 | 0.300 | 0.313 | 0.353 | 0.538 | 0.630 |
| | ✓ | 0.406 | 0.180 | 0.330 | 0.250 | 0.393 | 0.553 | 0.730 |
| **METIS (Ours)** | ✓ | 0.410 | 0.210 | 0.310 | 0.250 | 0.390 | 0.558 | 0.740 |

*Table 12.* Llama-3.2-3B per-task and average results.

| Method | Many-shot | Avg. | Instruction | Math | Multilingual Understanding | | | Safety |
|---|---|---|---|---|---|---|---|---|
| | | | IFEval | GSM8K | M-MMLU | M-ARC | M-HellaSwag | XSTest |
| Task Arithmetic | ✗ | 0.303 | 0.140 | 0.150 | 0.243 | 0.315 | 0.420 | 0.550 |
| | ✓ | 0.370 | 0.090 | 0.290 | 0.513 | 0.388 | 0.513 | 0.430 |
| DARE | ✗ | 0.348 | 0.130 | 0.240 | 0.313 | 0.358 | 0.498 | 0.550 |
| | ✓ | 0.384 | 0.190 | 0.330 | 0.375 | 0.388 | 0.540 | 0.480 |
| TIES | ✗ | 0.381 | 0.090 | 0.350 | 0.520 | 0.388 | 0.558 | 0.380 |
| | ✓ | 0.395 | 0.190 | 0.360 | 0.338 | 0.393 | 0.533 | 0.560 |
| ConsensusTA | ✗ | 0.398 | 0.210 | 0.250 | 0.390 | 0.380 | 0.535 | 0.620 |
| | ✓ | 0.395 | 0.220 | 0.340 | 0.343 | 0.370 | 0.528 | 0.570 |
| **METIS (Ours)** | ✓ | 0.425 | 0.220 | 0.340 | 0.378 | 0.468 | 0.533 | 0.610 |

*Table 13.* Llama-3.1-8B per-task and average results.

| Method | Many-shot | Avg. | Instruction | Math | Multilingual Understanding | | | Safety |
|---|---|---|---|---|---|---|---|---|
| | | | IFEval | GSM8K | M-MMLU | M-ARC | M-HellaSwag | XSTest |
| Task Arithmetic | × | 0.435 | 0.160 | 0.310 | 0.397 | 0.413 | 0.540 | 0.790 |
| | ✓ | 0.466 | 0.100 | 0.570 | 0.595 | 0.493 | 0.598 | 0.440 |
| DARE | × | 0.512 | 0.260 | 0.410 | 0.475 | 0.440 | 0.588 | 0.900 |
| | ✓ | 0.512 | 0.180 | 0.570 | 0.570 | 0.458 | 0.613 | 0.680 |
| TIES | × | 0.510 | 0.160 | 0.600 | 0.578 | 0.490 | 0.633 | 0.600 |
| | ✓ | 0.553 | 0.160 | 0.600 | 0.578 | 0.490 | 0.633 | 0.860 |
| ConsensusTA | × | 0.435 | 0.120 | 0.270 | 0.405 | 0.423 | 0.550 | 0.840 |
| | ✓ | 0.547 | 0.210 | 0.640 | 0.495 | 0.475 | 0.630 | 0.830 |
| **METIS (Ours)** | ✓ | 0.566 | 0.240 | 0.670 | 0.648 | 0.448 | 0.550 | 0.840 |

*Table 14.* Qwen-3-4B per-task and average results.

| Method | Many-shot | Avg. | Instruction | Math | Multilingual Understanding | | | Safety |
|---|---|---|---|---|---|---|---|---|
| | | | IFEval | GSM8K | M-MMLU | M-ARC | M-HellaSwag | XSTest |
| Task Arithmetic | × | 0.559 | 0.200 | 0.660 | 0.600 | 0.460 | 0.553 | 0.880 |
| | ✓ | 0.604 | 0.150 | 0.860 | 0.645 | 0.485 | 0.533 | 0.950 |
| DARE | × | 0.524 | 0.180 | 0.540 | 0.603 | 0.423 | 0.528 | 0.870 |
| | ✓ | 0.613 | 0.250 | 0.780 | 0.675 | 0.470 | 0.553 | 0.950 |
| TIES | × | 0.586 | 0.250 | 0.780 | 0.590 | 0.485 | 0.520 | 0.890 |
| | ✓ | 0.580 | 0.220 | 0.720 | 0.630 | 0.463 | 0.535 | 0.910 |
| ConsensusTA | × | 0.564 | 0.220 | 0.660 | 0.597 | 0.468 | 0.550 | 0.890 |
| | ✓ | 0.594 | 0.260 | 0.730 | 0.645 | 0.453 | 0.545 | 0.930 |
| **METIS (Ours)** | ✓ | 0.620 | 0.290 | 0.810 | 0.653 | 0.458 | 0.548 | 0.960 |

