# OpenReview forum: "Post-Hoc Merging is Not Enough: Many-Shot Model Merging with Loss-Gap Balancing"
_ICML.cc/2026/Conference — ICML 2026 regular_

### Official Review · Reviewer_2iBG · 2026-03-08

**Soundness:** 3
**Presentation:** 2
**Significance:** 3
**Originality:** 2
**Overall Recommendation:** 4
**Confidence:** 4

**Summary:**

This paper studies the problem of merging multiple task-specific large language models into a single multi-task model without joint multi-task training. The authors observe that most existing model merging methods perform a one-shot post-hoc aggregation after task-specific fine-tuning, which can lead to task interference and information erasure. To address this issue, the paper proposes a many-shot merging framework that iteratively alternates between task-specific local updates and parameter merging, allowing the merged model to gradually adapt to cross-task interactions. Building on this framework, the authors introduce METIS, a merging method that incorporates loss-gap-aware weighting to balance task contributions and a consensus-based masking mechanism to localize task-specific updates. Experiments on several LLM backbones (e.g., Gemma-2-2B, Llama-3.2-3B, and Llama-3.1-8B) across multiple task categories demonstrate that the proposed approach improves overall multi-task performance and achieves better worst-task robustness compared with several existing model merging baselines.

**Compliance With Llm Reviewing Policy:**

Affirmed.

**Final Justification:**

The rebuttal provides additional experiments that clarify the contributions of many-shot merging and the proposed components. While this helps strengthen the empirical support, it does not significantly change my overall assessment, as the main contribution still appears to be a combination of existing ideas with moderate novelty.
My evaluation remains unchanged.

**Key Questions For Authors:**

1. Clarification of results reported in Table 2.

In Table 2, METIS is reported with identical values under both the post-hoc and many-shot columns. Since METIS is described as a many-shot merging method, it is unclear how the post-hoc results are obtained and why the values appear identical. Could the authors clarify whether METIS is actually evaluated under a post-hoc setting, or whether these numbers correspond only to the many-shot configuration? Clarifying this would improve the interpretability of the comparisons and ensure a fair understanding of the experimental setup.

2. Computational cost of many-shot merging.

The proposed framework requires repeated rounds of task-specific local updates across R merging rounds. This likely introduces additional training cost compared to standard post-hoc merging, which typically requires only a single round of task-specific training. Could the authors provide runtime statistics (e.g., training time or GPU hours) comparing many-shot merging with standard post-hoc merging? Understanding this cost-performance trade-off would help evaluate the practical applicability of the method.

3. Source of improvement: iterative merging vs. repeated local training.

The many-shot framework alternates between local training and merging. It would be helpful to understand how much of the improvement comes from the iterative merging process itself versus repeated task-specific fine-tuning. For example, have the authors considered evaluating a variant that performs multiple merging steps without additional local training updates? Such a comparison could help isolate the contribution of iterative merging from that of additional training.

4. Stability of many-shot improvements across tasks.

While many-shot merging generally improves the average performance across methods, Table 2 suggests that performance on certain task categories (e.g., safety for some baselines on Llama-3.2-3B) may decrease after applying many-shot merging. Could the authors comment on whether these trade-offs are expected and whether METIS is designed to explicitly control such task-level performance shifts? Understanding this behavior would help clarify the robustness of the proposed approach.

**Limitations:**

The paper does not explicitly include a dedicated discussion of limitations or broader societal impacts. While the empirical results are extensive, it would be helpful for the authors to discuss the practical limitations of the proposed many-shot framework, particularly the additional computational cost introduced by repeated rounds of task-specific training. The authors could also comment on how the approach might scale when merging a larger number of tasks or models in more heterogeneous settings.

**Strengths And Weaknesses:**

Strengths

Soundness:The paper is generally technically sound. The method is clearly formulated, and the design of METIS is coherent: the many-shot merging framework is combined with loss-gap-aware weighting and consensus-based masking in a logically consistent way. The ablation study also supports the claim that these components contribute jointly to the final performance.

Presentation:The paper is overall clearly written and easy to follow. The motivation is well explained, especially the discussion of task interference and information erasure in one-shot post-hoc merging. The method section presents the framework in a relatively structured way, and the experimental section covers several complementary evaluations, including average performance, per-task performance, and worst-task robustness.

Significance:The paper studies an important and practical problem: how to combine multiple task-specific models into a single multi-task LLM without expensive joint multi-task training. This is a relevant setting for model merging research and for practical deployment, and improvements in this direction could be useful for building stronger multi-task systems at lower training cost than full joint training.

Originality:While the individual ingredients are related to existing ideas, the paper provides a meaningful combination of iterative many-shot merging, task-aware weighting, and masking. In addition, the paper highlights an interesting empirical insight: simply moving from post-hoc merging to iterative many-shot merging can already improve multi-task performance in many settings. This perspective itself is useful and contributes to understanding model merging behavior.

Weaknesses

Soundness:Although the empirical results are generally supportive, some claims appear slightly stronger than what the results fully establish. In particular, the benefits of many-shot merging are not uniform across all methods and task categories. For example, in Table 2 on Llama-3.2-3B, safety performance drops for some baselines after applying many-shot merging, even if the average performance improves. This suggests that the gains may partly come from task rebalancing rather than consistent improvement across all tasks.

Presentation:Some result presentation is confusing and would benefit from clarification. In Table 2, METIS is reported with identical values under both the post-hoc and many-shot columns, even though METIS is presented as a many-shot method. This makes the comparison somewhat difficult to interpret. It would also help to clarify how the normalized results in different figures and tables relate to each other.

Significance:The many-shot framework requires repeated rounds of task-specific local training, which likely increases computational cost substantially compared with standard post-hoc merging. However, the paper does not provide runtime, training-time, or GPU-cost analysis. Since efficiency is one of the main motivations for model merging, the lack of cost discussion weakens the practical significance of the proposed framework.

Originality:The conceptual novelty of the overall framework may be moderate. The iterative local-update-and-aggregation procedure resembles strategies from distributed or federated optimization, and the ablation results suggest that a substantial portion of the improvement may come from the many-shot protocol itself. While the loss-gap-aware weighting is a reasonable addition, the paper could more clearly distinguish what is fundamentally new in METIS beyond adopting an iterative merging scheme.

---

> ### Author Rebuttal · Authors · 2026-03-31
>
> We thank Reviewer 2iBG for the insightful comments. Below, we address the raised concerns.
>
> ---
> # 1. (Q2&W3) Runtime and Compute Cost Comparison
>
> **To provide a clear comparison, we measure the training-time with Llama-3.2-3B on our simulation setting**. The results are shown below.
>
> |Method|Training Time (s)|Avg. Performance|
> |-|-|-|
> |TA	|883.48| 0.857|
> |DARE|874.89| 0.914|
> |TIES|900.58| 0.938|
> |ConsensusTA|904.79| 0.945|
> |METIS (Ours)|918.53|**1.015**|
>
> This shows that METIS incurs a training-time overhead compared to post-hoc baselines. However, the total amount of task-specific training is comparable. post-hoc trains each task for R epochs, whereas our method performs 1 epoch over R rounds. Thus, the overhead mainly comes from repeated merging rather than additional training. This results in an increase in training time, which **we find to be a reasonable trade-off given the consistent improvements in multi-task performance and robustness**.
>
> ---
> # 2. (Q3) Source of Improvement
>
> To analyze the source of improvement (iterative merging vs. repeated local training), we perform an additional experiment that isolates these effects with Llama-3.2-3B model on the same simulation setting as our paper.
> |Model | Avg. Performance|
> |-|-|
> | Many-shot after Training | 0.966 |
> | Many-shot during Training (METIS)| 1.015 |
>
> - **Many-shot after Training**: perform all task-specific updates for R steps independently, and then apply R rounds of merging without further local updates.
> - **Many-shot during Training (METIS)**: interleave 1-step local updates and merging at each round, repeated for R rounds.
> - Both settings use the same total number of training steps, differing only in when merging is performed.
>
> As shown in the table, 'Many-shot during Training (METIS)' outperforms 'Many-shot after Training'. This indicates that the performance gain is not due to additional training, but rather **comes from the iterative integration of task-specific updates during training**, which enables the model to progressively adapt to cross-task interactions.
>
> Furthermore, **our method does not introduce additional local training beyond what is used in standard post-hoc merging**. Instead, the key difference lies in how these same updates are scheduled and integrated. Therefore, **the total number of local training steps per task remains identical across both settings**, ensuring a fair comparison in terms of training budget.
>
> ---
> # 3. (Q4&W1) Stability of Many-shot Improvements across Tasks
>
> We understand the reviewer’s concern regarding potential task-level trade-offs. To clarify this behavior, **we present the worst-task performance analysis** with Llama-3.2-3B backbone across IFEval, GSM8K, M-MMLU, M-ARC, M-HellaSwag, and XSTest data.
> |Method	| Avg |Worst Task|Gap|
> |-|-|-|-|
> |Task Arithmetic|0.706|0.385|-0.321|
> |DARE|0.807|0.542|-0.265|
> |TIES|0.883|0.375|-0.508|
> |ConsensusTA|0.942|0.641|-0.301|
> |**METIS (ours)**|**1.015**|**0.872**|**-0.143**|
>
> As shown above, while baseline methods exhibit a large gap between average and worst-task performance (indicating task-level imbalance), METIS significantly reduces this gap while also improving the worst-task performance. **This suggests that the gains are not due to sacrificing certain tasks, but rather from more balanced integration across tasks, leading to improved robustness.**
>
> ---
> # 4. (W4) Is METIS More Than Many-Shot Merging?
>
> To clarify the distinct contribution of METIS beyond many-shot merging, **we provide additional ablation results across multiple model scales, as shown below.**
>
> |Model | w/o loss gap | w/ loss gap|
> |--|--|--|
> |Gemma-2B| 0.688 | **0.800** |
> |Llama-3B| 0.927 | **1.015** |
> |Llama-8B| 0.869 | **0.935** |
>
> These results show that **loss-gap-aware weighting is not a marginal addition but a critical component that consistently improves performance across backbones**. While many-shot merging improves the optimization process, it does not explicitly address task imbalance. In contrast, loss-gap weighting dynamically prioritizes underperforming tasks, effectively mitigating information erasure.
>
> ---
> # 5. (Q1&W2) Clarifying METIS Placement in Table 2
>
> The identical values for METIS in both the post-hoc and many-shot columns were intentionally presented to enable a direct and intuitive comparison with existing post-hoc merging methods. **Since METIS is inherently a many-shot merging approach, its results are obtained under the many-shot setting, and the same values were duplicated in the post-hoc column for direct comparison.**
>
> To avoid potential confusion, we have revised Table 2. The corrected full table is available at https://anonymous-projectpage6.github.io/Loss-Gap-Aware-Many-Shot-Model-Merging/.
>
> ---
> We hope that our clarifications regarding the experimental setup, computational cost, and the role of many-shot merging have addressed your concerns. If any issues remain, we would be happy to provide further clarification.

---

> > ### Author Rebuttal · Reviewer_2iBG · 2026-04-03
> >
> > Thank you for the detailed rebuttal. The rebuttal clarifies the computational cost and provides useful evidence that the gains are not due to additional training.
> > However, it remains somewhat unclear how much of the improvement should be attributed to the many-shot protocol itself versus the specific METIS components.

---

> > > ### Author Response · Authors · 2026-04-04
> > >
> > > Thank you for the thoughtful follow-up. We appreciate the reviewer’s request and are glad to clarify how much of the improvement should be attributed to the many-shot protocol itself versus the specific METIS components.
> > >
> > > Our method consists of two parts: the many-shot merging framework and loss-aware consensus. **To explicitly isolate their contributions, we build on the component-wise ablation study presented in the original paper (Section 5.3, Figure 4), and extend it by additionally including a many-shot-only configuration for clearer comparison.** The results below not only include the Llama-3.2-3B backbone, which was already reported in the original paper, but also incorporate **additional experiments on different backbone models (Llama-3.1-8B and Gemma-2-2B)**, providing consistent evidence across various backbone architectures and reinforcing the generality of our findings.
> > >
> > > |Model|Many-shot | Loss-aware Consensus | Norm. Performance |
> > > |-|-|-|-|
> > > Llama-3.2-3B|V|X| 0.911|
> > > ||X|V| 0.924|
> > > ||V|V| **1.015**|
> > > Llama-3.1-8B|V|X| 0.885|
> > > ||X|V| 0.915|
> > > ||V|V| **0.935**|
> > > Gemma-2-2B|V|X| 0.703|
> > > ||X|V| 0.765|
> > > ||V|V| **0.800**|
> > >
> > > The results clearly disentangle the contributions of the many-shot protocol and the METIS components.
> > > - **The proposed components are effective even without many-shot**: Applying loss-aware consensus without the many-shot framework already surpasses the many-shot-only configuration, showing that the gains are not solely driven by iterative merging.
> > > - **The full method achieves the best performance**:
> > > Only when all components are combined do we observe the largest gain, demonstrating that the overall improvement arises from their complementary interaction rather than any single factor.
> > >
> > > Overall, we hope these results provide a clearer picture regarding the reviewer’s concern. The results indicate that **the improvement cannot be attributed to the many-shot protocol alone, but instead arises from the combined and complementary effects of the proposed components.**
> > >
> > > If there are any remaining questions or concerns, we would be happy to further discuss and clarify them. We hope that our clarifications have sufficiently addressed the reviewer’s concerns and would greatly appreciate your reconsideration on your final score.

---

### Official Review · Reviewer_iTwM · 2026-03-11

**Soundness:** 3
**Presentation:** 3
**Significance:** 3
**Originality:** 2
**Overall Recommendation:** 4
**Confidence:** 2

**Summary:**

This paper studies iterative many-shot model merging as an alternative to conventional post-hoc merging for building multi-task models from task-specialized checkpoints. Motivated by theoretical and empirical evidence that one-shot post-hoc merging suffers from task interference and information erasure, the authors propose METIS, a loss-aware many-shot merging framework. The method introduces consensus-based parameter masking and loss-gap-guided task-aware weighting to better balance task contributions during iterative merging. Experimental results across multiple model backbones show that the proposed approach consistently outperforms existing merging baselines on multi-task performance.

**Compliance With Llm Reviewing Policy:**

Affirmed.

**Final Justification:**

The authors’ response has addressed my concerns, and I am happy to maintain my positive score.

**Key Questions For Authors:**

1. The ConsensusTA baseline shows consistently strong performance on the instruction task across different model backbones and experimental settings, except for the Llama-3.1-8B model under the post-hoc merging protocol. A similar trend can also be observed on the math task. Could the authors provide further analysis or intuition to explain this behavior? In particular, it would be helpful to understand under what conditions ConsensusTA is expected to perform strongly or fail.

2. The experiments mainly focus on the LLaMA family of models. Could the authors report or discuss the performance of the proposed method on the Qwen model family? Such results would help better assess the generality of the proposed approach across different pretraining recipes and model architectures.

**Limitations:**

yes

**Strengths And Weaknesses:**

**Strengthes**
1. Model merging is an important and timely research topic, and the motivation behind the proposed method is well justified.
2. The theoretical analysis is clearly presented and generally well organized.

**Weaknesses**
1. The experimental comparison appears to miss several relevant baselines. As mentioned in the second paragraph of the related work section, there exist prior approaches that go beyond conventional post-hoc model merging. However, these methods are not included in the empirical evaluation, which makes it difficult to fully assess the relative advantages of the proposed approach.

2. Figure 1 may be somewhat misleading due to the use of performance normalization and the truncation of the vertical axis, which could exaggerate the perceived performance gap. Presenting the results with a consistent scale or providing the raw performance values would improve clarity and transparency.

3. Table 2 is difficult to read. Using bold formatting for both the best and second-best results reduces readability. A more standard presentation would be to highlight the best result in bold and the second-best result with an underline.

---

> ### Author Rebuttal · Authors · 2026-03-31
>
> We sincerely thank Reviewer iTwM for the thoughtful and constructive feedback. Below, we address the reviewer’s concerns.
>
> ---
> # 1. (Q2) Performance Report of the Qwen Model
> As already presented in Figure 6, we have applied our proposed method to the Qwen model family. For clarity, we reorganize those results into a tabular format below.
>
> | Qwen | Avg   | IFEval | GSM8k | M-MMLU | M-ARC | M-HellaSwag | XSTest |
> | - | -|- | - | - | - | - | - |
> | 0.6B | 0.379 | 0.160| 0.440 | 0.390  | 0.313 | 0.323| 0.650  |
> | 1.7B | 0.472 | 0.200| 0.530 | 0.498  | 0.355 | 0.430| 0.820  |
> | 4B   | 0.620 | 0.290| 0.810 | 0.653  | 0.458 | 0.548 | 0.960  |
> | 8B   | 0.594 | 0.320| 0.640 | 0.695  | 0.498 | 0.623| 0.790  |
> | 14B  | 0.629 | 0.310| 0.770 | 0.700  | 0.508 | 0.645| 0.840  |
>
> These results demonstrate that our method achieves strong and consistent performance across the Qwen family, supporting its generalizability beyond a specific model architecture.
>
> ---
> # 2. (W1) Performance Report of Relevant Baselines
>
> We agree that including recent iterative (i.e., many-shot) merging baselines would strengthen the comparison. Accordingly, **we perform additional experiments with recent iterative merging approaches [1], [2]** on the Llama-3B backbone and the same simulation setting as our paper.
> |Method|Category|Performance|
> |-|-|-|
> |**METIS (ours)**|Iterative|**1.015**|
> |FedMerge[1]||0.928|
> |ColD Fusion[2]||0.924|
> |ConsensusTA|Post-hoc|0.942|
> |TIES||0.883|
> |DARE||0.807|
> |TA||0.706|
>
> These results suggest that **iterative merging generally outperforms most conventional post-hoc methods**, highlighting the benefit of progressively integrating task updates. At the same time, **our method achieves the best performance overall**, demonstrating that combining iterative merging with loss-gap-aware weighting further improves multi-task integration beyond existing approaches.
>
> - [1] "FedMerge: Federated Personalization via Model Merging," AAAI 2025.
> - [2]  "ColD Fusion: Collaborative Descent for Distributed Multitask Finetuning," ACL 2023.
>
> ---
> # 3. (Q1) Behavior of ConsensusTA
>
> To better understand under what conditions ConsensusTA performs strongly or degrades, we revisit its performance across backbones and merging settings. The average normalized performance is summarized below.
> |Backbone|Post-hoc |Many-shot|
> |-|-|-|
> |Gemma-2-2B|0.752|0.791|
> |Llama-3.2-3B|0.942|0.945|
> |Llama-3.1-8B|0.694|0.898|
>
> ConsensusTA performs strongly in most settings, but shows a drop for Llama-3.1-8B under post-hoc merging, while recovering under many-shot merging.
>
> We hypothesize that this behavior arises from the **interaction between model scale and merging strategy**. As model size increases, task-specific updates tend to become more diverse, reducing overlap across tasks. Under post-hoc merging, this reduced overlap led to performance degradation. In contrast, many-shot merging progressively integrates task updates, which may alleviate this issue and lead to more stable performance
>
> ---
> # 4. (W2) Clarification of Figure 1 and Use of Normalized Performance
>
> We appreciate the reviewer’s comment regarding the clarity of Figure 1. To improve transparency, we present the same results in a tabular form below, without relying on axis scaling.
> |Method| Gemma-2-2B | | Llama-3.2-3B ||
> |-|-|-|-|-|
> ||Post-hoc |Many-shot |Post-hoc |Many-shot |
> |Task Arithmetic | 0.521 | 0.715 | 0.706 | 0.857|
> |DARE| 0.663 | 0.701| 0.807 | 0.914 |
> |TIES| 0.726 | 0.776| 0.883 | 0.938 |
> |ConsensusTA| 0.752 | 0.791 | 0.942 | 0.945 |
>
> Across both Gemma-2-2B and Llama-3.2-3B, we consistently observe that transitioning from post-hoc to many-shot merging improves performance across all methods. **We report normalized performance to account for differences in task difficulty and scale across heterogeneous benchmarks, following standard practice in prior work [3–7]**. In such settings, absolute performance values are not directly comparable across tasks, and normalized metrics provide a more balanced and interpretable comparison. For completeness, **we note that the raw performance values are already reported in the appendix (Tables 8–10).**
>
> - [3] “MergeBench: A Benchmark for Merging Domain-Specialized LLMs,” NeurIPS 2025.
> - [4] “Localizing Task Information for Improved Model Merging and Compression,” ICML 2024.
> - [5] “TIES-Merging: Resolving Interference When Merging Models,” NeurIPS 2023.
> - [6] “Model Soups: Averaging Weights of Multiple Fine-Tuned Models Improves Accuracy Without Increasing Inference Time,” ICML 2022.
> - [7] “Merge-of-Thought Distillation,” arXiv:2509.08814.
>
> ---
> # 5. (W3) Improved Table Readability and Formatting
>
> We have revised Table 2 to follow the standard convention to improve readability. The revised full table is available at https://anonymous-projectpage6.github.io/Loss-Gap-Aware-Many-Shot-Model-Merging/.
>
> ---
> We hope that our additional clarifications further strengthen the paper. If any questions remain, we would be happy to discuss them.

---

> > ### Author Rebuttal · Reviewer_iTwM · 2026-04-04
> >
> > The authors’ response has addressed my concerns, and I am happy to maintain my positive score.

---

> > > ### Author Response · Authors · 2026-04-06
> > >
> > > We sincerely thank Reviewer iTwM for the careful reading and for the positive assessment. We especially appreciate that the reviewer recognized the strong motivation of the work, as well as the clear and well-organized presentation of the theoretical analysis and the consistent empirical validation across multiple model backbones.
> > >
> > > We are glad that our rebuttal has successfully addressed the main concerns and questions raised in the original review. In particular, we have **(W1)** incorporated additional relevant baselines, including iterative merging approaches, for a more comprehensive comparison, **(W2)** clarified the presentation of Figure 1 by providing tabular results without normalization effects, and **(W3)** improved the readability of Table 2 following standard conventions. In addition, we also have **(Q1)** provided further analysis on the behavior of ConsensusTA across different backbones and merging settings, and **(Q2)** included additional results on the Qwen model family to demonstrate generality.
> > >
> > > **Our key contribution is to show that task interference and information erasure in post-hoc merging can be understood through accumulated model drift, and that this can be effectively mitigated through iterative many-shot merging with loss-gap-aware weighting**. Building on the reviewer’s positive assessment of the motivation and theoretical analysis, we provide both theoretical and empirical evidence supporting this perspective.
> > >
> > > Based on this insight, the proposed framework enables more balanced integration of task updates across different backbones, leading to consistent improvements in multi-task performance and demonstrating the effectiveness and generality of the approach.
> > >
> > > We sincerely appreciate the reviewer’s thoughtful evaluation, and if the reviewer finds that these clarifications strengthen the overall contribution and significance of the work, we would be grateful for your consideration in the final scoring. Given that the concerns have been fully addressed and the scores are consistently positive across reviewers, we believe the work is currently at the acceptance boundary. In this situation, the reviewer’s final recommendation could be especially decisive in the overall decision, and we would greatly appreciate your consideration in the final scoring.

---

### Official Review · Reviewer_RLdc · 2026-03-12

**Soundness:** 2
**Presentation:** 3
**Significance:** 2
**Originality:** 2
**Overall Recommendation:** 4
**Confidence:** 4

**Summary:**

This paper argues that one-shot post-hoc merging is intrinsically suboptimal for multi-task LLM composition, and that simply switching to an iterative “many-shot” merging protocol already improves performance. On top of that, it proposes METIS, which combines task-wise loss-gap reweighting with consensus-based masking inside the many-shot framework. The evaluation uses four task categories (instruction-following, math, multilingual, safety), different LLM architectures and model sizes. The main claim is that METIS improves average normalized performance, worst-task robustness, and retention of pre-trained knowledge relative to standard merging baselines.

**Compliance With Llm Reviewing Policy:**

Affirmed.

**Final Justification:**

The rebuttal has fully addressed my concerns. While the method of many-shot merging is an interessting paradigm for model merging, the initial revision lacked a comprehensive comparison against recent state-of-the-art methods and compared against weaker baselines. In addition, the original revision only evaluated on a small subset of tasks, questioning the validity of the method's performance. Furthermore, the authors did not provide initial justification about the variation in performance improvements for their method.

Nevertheless, the authors diligently and comprehensively addressed all of my concerns with additional results and experiments, justifying my score increase to a Weak Accept. The discussion and questions can be seen in the discussion and rebuttal below. With the additional experiments, the revision is now acceptance-worthy and I believe there is value in their paper being accepted.

**Key Questions For Authors:**

**Q1:**

 Instead of utilizing masking, have the authors thought about using orthogonalization and SVD in their methods, following [1], [2], and [3]? If so, could they please compare it to masking?

**Q2:**
Is there a reason mostly only normalized performance is shown? It would be useful to show the absolute performance as well as normalized performance side-by-side. Namely, Table 2. would significantly benefit from this.

**Q3:**
Table 2 shows METIS under both post-hoc and many-shot merging. This could confuse the reader into thinking that METIS is also a post-hoc method. Could the authors correct me if I misunderstood this table?

**References**:

[1] Marczak, D., Magistri, S., Cygert, S., Twardowski, B., Bagdanov, A. D., and van de Weijer, J. No task left behind: Isotropic model merging with common and task-specific subspaces. In International Conference on Machine Learning, 2025.

[2] Gargiulo, A. A., Crisostomi, D., Bucarelli, M. S., Scardapane, S., Silvestri, F., and Rodolà, E. Task singular vectors: Reducing task interference in model merging. In Conference on Computer Vision and Pattern Recognition, 2025.

[3] Skorobogat, R., Roth, K., and Georgescu, M. I. Subspace-boosted model merging. In arXiv preprint arXiv:2506.16506, 2025.

**Limitations:**

yes

**Strengths And Weaknesses:**

### **Strengths**

**1. Clear presentation and writing:**

The paper is well-written, easy to grasp and the figures are informative and intuitive. Additionally, the provided codebase helped to understand the method better and prevent any misunderstanding or explanations, omitted in the main paper.

**2. Introduction of Iterative Merging:**

The paper discusses an interesting paradigm for model merging, done iteratively instead of once in a post-hoc manner, which is the current standard. The authors show that many-shot merging is effective and can be applied to further improve popular baseline model merging methods.

**3. Application to Timely Domains and Architectures:**

The paper applies METIS to timely domains such as Instruction tuning, multilingual knowledge, and safety, which are generally underexplored in the  model merging field. In addition, the authors showcase the effectiveness of their method utilizing a range of LLM models, spanning from 2B to 8B sizes.



### **Weaknesses**

**1. Low Number of Tested Tasks**

While the authors apply their method to timely domains, the number of merged task groups is only 4, subdivided into 6 datasets.
It would be beneficial to evaluate results for an increasing number of tasks, since model merging performance degrades significantly as more tasks are merged. A more thorough analysis from this perspective would be highly informative. For example, it is common to evaluate a method using 8, 14 and 20 tasks, respectively, for CV tasks and 7 to 8 tasks for NLP tasks, as presented here [1], whose methods the authors have used (ConsensusTA).

**2. Insufficient Baseline Comparison**:

The authors show the benefit of their approach compared to widely used methods such as Task Arithmetic, DARE, TIES and ConsensusTA. Nevertheless, they omit key state-of-the-art baslines such as TSV-M [3], Iso-C, Iso-CTS [2], and Subspace Boosting [3]. The authors should benchmark their method against these methods, since all three significantly outperform the baselines in the paper. It is unclear to me whether METIS would be competitive under this comparison, only slightly outperforming ConsensusTA, from what I can see.

**3. Post-hoc Vs. Many-Shot Merging**:

Following the previous point, it is unclear to me how well many-shot merging would actually perform when utilizing the new baselines from W.2. For example, for ConsensusTA, it seems that Many-shot merging is basically equivalent to post-hoc merging. Could the authors please elaborate on this point and explain the reason why many-shot merging does not improve ConsensusTA? Similarly, Figure 4. seems to indicate the same conclusion about the ineffectiveness of many-shot merging in improving performance.

**4. Computational Overhead of METIS**:

It would be highly useful to show a computational overhead of METIS compared to other training-free and post-hoc methods [1], [2], [3], [4].

**References:**

[1] Wang, K., Dimitriadis, N., Ortiz-Jimenez, G., Fleuret, F., and Frossard, P. Localizing task information for improved model merging and compression. In International Conference on Machine Learning, 2024.

[2] Marczak, D., Magistri, S., Cygert, S., Twardowski, B., Bagdanov, A. D., and van de Weijer, J. No task left behind: Isotropic model merging with common and task-specific subspaces. In International Conference on Machine Learning, 2025.

[3] Gargiulo, A. A., Crisostomi, D., Bucarelli, M. S., Scardapane, S., Silvestri, F., and Rodolà, E. Task singular vectors: Reducing task interference in model merging. In Conference on Computer Vision and Pattern Recognition, 2025.

[4] Skorobogat, R., Roth, K., and Georgescu, M. I. Subspace-boosted model merging. In arXiv preprint arXiv:2506.16506, 2025.

---

> ### Author Rebuttal · Authors · 2026-03-31
>
> We appreciate the reviewer’s careful reading and valuable feedback. Below, we address the raised concerns.
>
> ---
> # 1. (W1) Additional Results with More Tasks
> We agree with the reviewer that evaluating model merging under a larger number of tasks is important, as performance is known to degrade with increasing task diversity. In response, **we extend our experiments to settings with 7 [1], 8 [2] and 11 [3] tasks**. The results are presented below.
>
> | Backbone | Method | 7 Tasks [1] | 8 Tasks [2] | 11 Tasks [3]|
> |--|--|--| --|--|
> |Llama 3B| TA |0.997|0.970| 1.021|
> ||DARE        |1.006|0.975| 0.812|
> ||TIES        |0.871|0.822| 0.976|
> ||ConsensusTA |0.927|0.823| 0.873|
> ||**METIS (ours)**|**1.015**|**0.993**| **1.036** |
> |Llama 8B|TA    |1.018|0.885|0.946 |
> ||DARE            |0.929|0.850|0.932 |
> ||TIES            |0.813|0.754| 0.844|
> ||ConsensusTA     |0.928|0.759| 0.835|
> ||**METIS (ours)**|**1.052**|**0.909**|**1.021** |
>
> The results show that **METIS consistently achieves strong performance across diverse datasets, demonstrating its effectiveness in handling multi-task interference**. We will include these extended experiments in the final version for completeness.
>
> ---
> # 2. (Q1&W2&W4) Comparison with recent SOTA
>
> **We perform additional experiments by applying many-shot merging to the four methods (TSV-M, Iso-C, Iso-CTS, and Subspace Boosting), and compare both performance and computational cost** measured by total training time under Llama-3B backbone and the same experimental setup as in the paper.
> |Method|Avg. Performance|Worst Task Performance|Training Time (s)|
> |-|-|-|-|
> | TSV-M  | 0.953|0.750|3219.07|
> | Iso-C   | 0.941|0.792|1050.18|
> | Iso-CTS | 0.920|0.653|1225.03|
> | TA+Subspace Boosting | 0.885 | 0.458 |1252.42|
> | **METIS (ours)**  | **1.015**|**0.872**|**918.53**|
>
> **The results show that METIS achieves the best overall performance and the highest worst-task performance among all methods, while also requiring the least total training time**, demonstrating both its effectiveness and efficiency compared to recent methods.
>
> The higher computational cost of prior methods mainly comes from their reliance on expensive operations such as SVD, subspace decomposition, or orthogonalization during merging. **In contrast, METIS only uses lightweight element-wise operations, resulting in lower total training time.**
>
> ---
> # 3. (W3) Post-hoc Vs. Many-shot merging
>
> To further analyze the reviewer’s concern regarding the limited improvement of many-shot merging for ConsensusTA, **we revisit its performance across multiple backbones and additionally include results on the Qwen-4B model under the same experimental setup as in our paper**. The normalized performance of ConsensusTA is summarized below.
> | Model        | Post-hoc | Many-shot | Improvement |
> | -| -| - | - |
> | Gemma-2-2B   | 0.752| 0.791| +3.9%|
> | Llama-3.2-3B | 0.942| 0.945| +0.3%|
> | Qwen-4B       | 1.046| 1.101| +5.5%|
> | Llama-3.1-8B | 0.694| 0.898| +20.4% |
>
> **We observe that many-shot merging consistently improves the performance of ConsensusTA across all evaluated models**. While the magnitude of improvement varies, and is relatively small for Llama-3.2-3B, other models exhibit clear and sometimes substantial gains. These results indicate that the effectiveness of many-shot merging is not limited to specific architectures but generalizes across diverse model families.
>
> ---
> # 4. (Q2&Q3) Table 2 improvement
>
> **We report normalized performance to account for differences in task difficulty and scale across heterogeneous benchmarks, following standard practice in prior work [1–5].** Since absolute values are not directly comparable across tasks, normalized metrics provide a more balanced evaluation. For completeness, **raw performance is already provided in the appendix (Tables 8–10)**.
>
> Regarding Table 2, METIS is inherently a many-shot method, and its results are obtained under the many-shot setting. **The identical values in both columns were included solely for direct comparison with post-hoc baselines, not to indicate that METIS operates in a post-hoc manner**. We agree with the reviewer that the presentation of Table 2 may confuse, and we have revised the table for clarity. **The revised full table is available at https://anonymous-projectpage6.github.io/Loss-Gap-Aware-Many-Shot-Model-Merging/**.
>
> - [1]  “TIES-Merging: Resolving Interference When Merging Models,” NeurIPS 2023.
> - [2] “Model Soups: Averaging Weights of Multiple Fine-Tuned Models Improves Accuracy Without Increasing Inference Time,” ICML 2022.
> - [3] “Localizing Task Information for Improved Model Merging and Compression,” ICML 2024.
> - [4] “MergeBench: A Benchmark for Merging Domain-Specialized LLMs,” NeurIPS 2025.
> - [5] “Merge-of-Thought Distillation,” arXiv:2509.08814.
>
> ---
> We hope that the additional experiments, extended comparisons, and clarifications have addressed your concerns, and we would greatly appreciate it if you could reconsider your evaluation and increase the score.

---

> > ### Author Rebuttal · Reviewer_RLdc · 2026-04-03
> >
> > I thank the authors for their detailed rebuttal. The additional results look very promising.
> >
> > Before I raise my score, could the authors please run Pt. 1 with the methods from Pt. 2? The results seem promising, but require running for more tasks once again.
> >
> > Secondly, could the authors provide more intuition and explanation for Pt. 3 why the 3B model sees such little improvement compared to 8B? Any additional explanation would be helpful.
> >
> > I thank the authors once again and am looking forward to their reply.

---

> > > ### Author Response · Authors · 2026-04-05
> > >
> > > We are glad that our previous rebuttal and additional experiments have successfully resolved the concerns and questions the reviewer raised, particularly regarding **(W1)** evaluating under a larger number of tasks, **(W2)** considering recent baselines, **(W4)** providing computational cost comparisons, **(Q1)** comparing both performance and efficiency, **(Q2)** clarifying the presentation of Table 2, **(Q3)** providing additional explanation of the normalized performance. We sincerely thank the reviewer for these thoughtful points, as they helped us strengthen our experimental validation and analysis.
> > >
> > > For your follow-up questions, we further perform and provide additional experiment results and analysis, as detailed below.
> > >
> > > ---
> > > # 1. Additional Results with More Tasks and Recent Baselines
> > >
> > > To address the reviewer’s suggestion, we conduct additional experiments including several recent baselines (TSV-M, Iso-C, Iso-CTS, and Subspace Boosting) under the many-shot merging protocol with expanded task settings (7, 8, and 11 tasks). This directly evaluates the scalability of the proposed approach under a larger number of tasks. **Bold** indicates the best performance and $\underline{\textrm{underline}}$ indicates the second-best performance.
> > >
> > > Model|Method|7 Tasks| 8 Tasks| 11 Tasks|
> > > |-|-|-|-|-|
> > > |Llama-3.2-3B|TSV-M|**1.023**|0.984|$\underline{1.030}$|
> > > ||Iso-C |0.960 |$\underline{0.985}$|1.025|
> > > ||Iso-CTS|0.947 |0.948 |1.003|
> > > ||TA +  Subspace Boosting|1.013|0.940|$\underline{1.030}$|
> > > ||METIS (Ours)|$\underline{1.015}$|**0.993**|**1.036**|
> > > |Llama-3.1-8B|TSV-M|$\underline{1.038}$|$\underline{0.897}$|1.002|
> > > ||Iso-C|0.986|0.892|1.006|
> > > ||Iso-CTS|0.911|0.830|0.852|
> > > ||TA +  Subspace Boosting|1.006|0.892|$\underline{1.014}$|
> > > ||METIS (Ours)|**1.052**|**0.909**|**1.021**|
> > >
> > >
> > > **Average Performance:** METIS maintains consistently strong performance across tasks. In particular, it achieves the best results in most configurations and remains competitive even in cases where other methods perform well (e.g., TSV-M in the 7-task setting).
> > >
> > > **Training time (s):** We further investigate the training time of TSV-M and METIS for fair comparison.
> > > Model|Method|7 Tasks|8 Tasks|11 Tasks|
> > > |-|-|-|-|-|
> > > |Llama-3.2-3B|TSV-M|898.673|1009.228|1169.304|
> > > ||METIS (Ours)|537.487|669.450|821.277|
> > > |Llama-3.1-8B|TSV-M|1701.056|2098.405|2497.602|
> > > ||METIS (Ours)|621.668|990.400|1394.891|
> > >
> > > METIS incurs lower training time in our experiments, relying only on lightweight operations that scale linearly with the number of tasks $T$ and parameter dimension $d$, i.e., $\mathcal{O}(Td)$. In contrast, TSV-M requires significantly higher training time due to its reliance on SVD, which leads to a computational complexity of approximately $\mathcal{O}(Td^3)$.
> > >
> > > Overall, the results suggest that METIS provides a better performance–efficiency trade-off.
> > >
> > > ---
> > > # 2. Analysis of Many-Shot Merging Behavior
> > >
> > > Q. Why does the 3B model see little improvement in many-shot compared to 8B?
> > >
> > > The 3B model shows modest changes in average performance because improvements are concentrated in weaker tasks, while stronger tasks are already near the ceiling. Many-shot merging primarily improves underperforming tasks, resulting in more balanced task-level performance rather than uniform gains across all tasks.
> > >
> > > To investigate this, we break down the average score into per-task performance. The results (from Table 2 of the original submission) are shown below.
> > >
> > >
> > > |Method | Avg. Performance | Instruction|Math|Multilingual|Safety|
> > > |-|-|-|-|-|-|
> > > |ConsensusTA |0.942|0.875|0.641|0.958|1.265|
> > > |ConsensusTA + Many-shot|0.945|0.917|0.872|0.907|1.163|
> > > |METIS (Ours) |1.015 |0.917|0.872|1.018|1.245|
> > >
> > >
> > > - **ConsensusTA** exhibits large variation across tasks, with strong tasks driving the average but weaker tasks underperforming.
> > > - **ConsensusTA + Many-shot** substantially improves weaker tasks (Math, Instruction), reducing task-level imbalance. Some tasks (Multilingual, Safety) slightly decrease, reflecting remaining cross-task trade-offs.
> > > - **METIS** maintains the gains in weaker tasks while largely preserving performance on other tasks, consistent with more stable and balanced multi-task integration.
> > >
> > > A similar pattern occurs for Llama-3.1-8B.
> > >
> > > |Method	|Avg. Performance|Instruction|Math	|Multilingual	|Safety|
> > > |-|-|-|-|-|-|
> > > |ConsensusTA|0.694|0.293|0.458|0.812|0.977|
> > > |ConsensusTA + Many-shot|0.898|0.512|1.085|0.943|0.965|
> > > |METIS (Ours)|0.935|0.585|1.136|0.972|0.977|
> > >
> > >
> > > While the magnitude of the average improvement differs, the underlying behavior remains consistent. Many-shot merging again improves weaker tasks and reduces performance imbalance across tasks, leading to a more balanced multi-task model. Furthermore, METIS achieves more balanced performance across all tasks.
> > >
> > > ---
> > > We sincerely thank the reviewer for this insightful question and suggestion. We hope that these additional results and explanations address your concerns, and we would greatly appreciate your reconsideration on final score.

---

### Official Review · Reviewer_XxLG · 2026-03-14

**Soundness:** 3
**Presentation:** 3
**Significance:** 2
**Originality:** 2
**Overall Recommendation:** 4
**Confidence:** 4

**Summary:**

This paper addresses the problem of task interference and information erasure in multi-task model merging. The authors argue that the dominant post-hoc merging paradigm — where task-specific models are merged only once after independent training — suffers from large model drift, leading to destructive cross-task interference during aggregation. To mitigate this, they propose replacing post-hoc merging with an iterative many-shot merging protocol, and further introduce METIS, which combines task-wise loss-gap-based weighting with consensus-based masking within this iterative framework. The paper makes three claimed contributions: (1) theoretical and empirical demonstration that many-shot merging reduces multi-task loss compared to post-hoc merging; (2) a loss-gap-aware task weighting mechanism that dynamically compensates for information erasure across rounds; (3) the METIS framework integrating the above with consensus-based masking. Experiments conducted on Gemma-2-2B, Llama-3.2-3B, and Llama-3.1-8B across four task categories show the effectiveness of the proposed METIS.

**Compliance With Llm Reviewing Policy:**

Affirmed.

**Final Justification:**

While the proposed method introduces additional constraints compared to standard model merging, transplanting the client drift perspective from federated learning into the merging framework offers an interesting angle, and I therefore lean toward acceptance.

**Key Questions For Authors:**

See weakness.

**Limitations:**

yes

**Strengths And Weaknesses:**

**Strengths**

- This paper draws on the concept of client drift from federated learning to reinterpret information erasure in model merging as a consequence of accumulated model drift, and uses this perspective to motivate the introduction of iterative aggregation into the merging framework. While many-shot merging is algorithmically equivalent to FedAvg, transplanting this analytical lens from federated learning to model merging offers a useful reframing of the problem, and the systematic empirical validation across multiple model families and merging methods provides reasonable support for its effectiveness.
- The paper also explicitly evaluates worst-task performance as a primary metric, rather than relying solely on average performance. METIS reduces the gap between average and worst-task performance from 0.51 (TIES) to 0.14 on Llama-3.2-3B, a substantial improvement. In practical deployment settings where severe degradation on any single task is often unacceptable, this evaluation perspective is well-motivated and worth adopting in future benchmarking efforts.

**Weaknesses**

- The three core components of this paper each have direct predecessors: the many-shot merging framework is equivalent to FedAvg and ColD Fusion; the loss-gap weighting mechanism closely mirrors ideas extensively explored in federated learning literature on client drift and heterogeneity-aware aggregation, such as q-FedAvg, none of which are cited or discussed in the paper; and consensus masking is directly adapted from ConsensusTA, with only incremental modifications. The combination of these components does not constitute a sufficiently novel contribution, and the authors should more clearly articulate what each component contributes beyond its respective predecessor.
- The fundamental appeal of model merging lies in the ability to train task-specific models entirely independently, preserving single-task performance and allowing flexible deployment — for instance, adding or replacing a task model without retraining others. However, many-shot merging requires all task models to be initialized from the same merged checkpoint at each round, introducing inter-task coupling that undermines this independence and contradicts the core motivation of the merging paradigm. In this setup, the training procedure becomes substantially similar to joint multi-task training, yet the paper includes no comparison against data-mixing baselines. Without this comparison, it is impossible to assess where METIS stands relative to the most natural multi-task learning alternative, and the practical value of the proposed approach remains difficult to evaluate.

---

> ### Author Rebuttal · Authors · 2026-03-31
>
> We thank the reviewer for the detailed and constructive feedback. We provide clarifications and additional evidence below to address the concerns.
>
> ---
> # 1. (W1-a)  How the METIS differs from prior FL methods
>
> We thank the reviewer for the insightful discussion on the connection between our method and prior federated learning approaches. **To clarify how METIS differs from prior FL methods, we perform experiments comparing it with representative approaches (ColD Fusion [1] and q-FedAvg [2])** on the Llama-3B backbone with the same data setup as in our paper. We will incorporate the relevant discussion and citations to prior work in both the related work and experimental sections of the paper. The aggregation rules of each method and the corresponding experimental results are summarized in the table below.
>
> |Method|Aggregation Rule|Avg. Performance|
> |-|-|-|
> |ColD Fusion [1]|$\Theta^{\texttt{r}} = \frac{1}{T}\sum_{\tau=1}^{T}  \theta_\tau^{\texttt{r}}$|0.911|
> |q-FedAvg [2]|$\Theta^{\texttt{r}} = \sum_{\tau=1}^{T} \frac{\mathcal{L}\_\tau(\Theta^{\texttt{r-1}})^q}{\sum\_{t \in T} \mathcal{L}\_t(\Theta^{\texttt{r-1}})^q} \theta\_\tau^{\texttt{r}}$|0.858|
> |**METIS(Ours)**|$\Theta^{\texttt{r}} = \sum_{\tau=1}^{T} \frac{\text{exp}(\mathcal{L}\_\tau(\Theta^{\texttt{r-1}})-\mathcal{L}\_\tau(\theta^{\texttt{r}}\_\tau))}{\sum\_{t \in T} \text{exp}(\mathcal{L}\_\tau(\Theta^{\texttt{r-1}})-\mathcal{L}\_\tau(\theta^{\texttt{r}}\_t))} \theta\_\tau^{\texttt{r}}$|**1.015**|
>
> **The results show that our method outperforms both ColD Fusion and q-FedAvg.** ColD Fusion applies uniform aggregation without considering task-specific effects, which limits its ability to handle task interference in multi-task merging. Interestingly, q-FedAvg, even though it uses loss-based reweighting, performs the worst. We believe this happens because it relies on absolute loss values. In a multi-task setting where different tasks have different loss scales, this can bias the aggregation toward tasks with larger losses. **In contrast, our method does not rely on raw loss values from a single round. Instead, it uses loss gaps, which better reflect cross-task interference and lead to more balanced integration.**
>
> We will incorporate this discussion into the main paper to more clearly position our method relative to prior FL approaches, and to explicitly highlight how loss-gap-based weighting differs from conventional loss-based aggregation.
>
> - [1] S. Don-Yehiya, et al., "ColD Fusion: Collaborative Descent for Distributed Multitask Finetuning," ACL 2023.
> - [2] T. Li, et al., "Fair Resource Allocation in Federated Learning," ICLR 2020.
>
> # 2. (W1-b) How the METIS differs from ConsensusTA
> While our consensus masking is closely related to that of ConsensusTA, **it is specifically designed to operate within an iterative many-shot merging process rather than a static aggregation scheme.** To directly illustrate this difference under a fair setting, we present the empirical comparison using the results in Tables 8–10 in Appendix D.
> |Model|Method|Avg. Performance| Worst Task Performance |
> |--|--|--|--|
> |Gemma-2B|ConsensusTA (many-shot)|0.791| 0.450|
> | |**METIS(Ours)**|**0.800**| **0.525**|
> |Llama-3B|ConsensusTA (many-shot)|0.945|0.825|
> | |**METIS(Ours)**|**1.015**|**0.872**|
> |Llama-8B|ConsensusTA (many-shot)|0.898|0.512|
> | |**METIS(Ours)**|**0.935**|**0.585**|
>
> From the results, we observe that **our method consistently outperforms ConsensusTA across all model architectures under the same many-shot setting, suggesting that adapting the masking mechanism to the iterative merging process leads to more stable and effective integration.**
>
> ---
> # 3. (W2) Comparison with Data-mixing Baseline
>
> To assess whether many-shot merging behaves similarly to joint multi-task training, **we perform additional experiments with data-mixing baseline TULU 3[3]** on the Llama-3B backbone across IFEval, GSM8K, M-MMLU, M-ARC, M-HellaSwag, and XSTest. The results are presented below.
> |Category|Method|Avg. Performance|
> |--|--|--|
> |Data-mixing|TULU 3[3]| 0.884 |
> |Many-shot merging|**METIS (ours)**|**1.015**|
>
> As shown in the table, our method outperforms TULU 3. **This indicates that many-shot merging is not equivalent to joint training, but instead provides a more effective way to integrate multiple tasks.** Rather than relying on fully coupled optimization over mixed data, our approach introduces controlled interaction across tasks, leading to more balanced and stable performance. Therefore, many-shot merging should be viewed as a distinct paradigm for multi-task integration, rather than a variant of joint multi-task training.
>
> [3] N. Lambert, et al., "Tulu 3: Pushing Frontiers in Open Language Model Post-Training," COLM 2025.
>
> ---
> We hope our clarifications regarding the novelty, many-shot merging, and its distinction from multi-task training address your concerns. We would appreciate your reconsideration.

---

> > ### Author Rebuttal · Reviewer_XxLG · 2026-04-03
> >
> > Thank you for your reply. My concerns are resolved, and I have decided to increase my score.

---

> > > ### Author Response · Authors · 2026-04-06
> > >
> > > Thank you very much for the thoughtful follow-up and for increasing the score. We truly appreciate the reviewer’s careful reading and constructive feedback.
> > >
> > > We are glad that our rebuttal has clearly addressed the main concerns, in particular **(W1)** the relationship to prior FL methods and the role of loss-gap weighting, and **(W2)** the additional comparisons and analysis, as well as the role of many-shot merging and its distinction from joint multi-task training.
> > >
> > > Beyond these clarifications, we would like to highlight the key strength of this work. Rather than introducing a single isolated technique, **our goal is to show that iterative integration combined with loss-gap-aware weighting provides a principled way to mitigate information erasure**, offering a different perspective from existing post-hoc merging approaches.
> > >
> > > **Importantly, this perspective translates into consistent improvements in worst-task performance and a significant reduction in the gap between average and worst-task performance.** As the reviewer also noted, this aspect is particularly important in practical deployment scenarios, where robustness across tasks is critical.
> > >
> > > We sincerely thank the reviewer again for the constructive suggestions, which helped us further strengthen the paper. If the reviewer finds that these clarifications enhance the overall contribution and significance of the work, we would greatly appreciate your consideration in the final assessment. Given that the concerns have been fully addressed and the current review outcomes appear to place the paper near the acceptance boundary, the reviewer’s final recommendation could play a particularly important role in the overall decision.

---

### Decision · Program_Chairs · 2026-04-30

**Decision:**

Accept (regular)

**Comment:**

This paper addresses the challenges of task interference and information erasure in multi-task model merging by proposing an iterative many-shot merging protocol as an alternative to one-shot post-hoc merging. The authors introduce the METIS framework, which combines a loss-gap-aware task weighting mechanism to dynamically compensate for information erasure across rounds with a consensus-based parameter masking mechanism. The paper presents three main contributions: a theoretical and empirical demonstration that many-shot merging reduces multi-task loss compared to post-hoc merging, the development of the loss-gap-aware task weighting mechanism, and the integration of these elements into the METIS framework. Experimental results across four task categories, including instruction-following, math, multilingual, and safety, using Gemma-2-2B, Llama-3.2-3B, and Llama-3.1-8B backbones demonstrate that METIS improves average normalized performance, worst-task robustness, and retention of pre-trained knowledge compared to existing baselines.

This paper receives all positive scores and is recommended for acceptance. Reviewers highlighted the value of adapting the client drift perspective from federated learning into the model merging context. Following the rebuttal, two reviewers adjusted their evaluations upwards. Reviewer RLdc initially noted issues with weak baselines, limited task evaluation, and a lack of justification for performance variations. After the authors diligently provided additional results and comprehensive experiments, Reviewer RLdc increased their score to a Weak Accept. Reviewer XxLG similarly increased their score upon having their concerns resolved. Reviewer iTwM was satisfied with the response and maintained their positive score. Reviewer 2iBG kept their evaluation unchanged, concluding that while the novelty is moderate due to combining existing ideas, the additional rebuttal experiments effectively clarified the contributions and strengthened the empirical support. Given the solid technique results and the thorough resolution of reviewer concerns, the paper merits acceptance.